# RDNAS: ROBUST DUAL-BRANCH NEURAL ARCHITECTURE SEARCH

## ABSTRACT

Deep neural networks achieve impressive accuracy yet remain highly suscepti-
ble to adversarial perturbations, limiting their deployment in safety-critical do-
mains. We propose **RDNAS**, a robust dual-branch neural architecture search
framework that jointly optimizes standard (clean) accuracy and adversarial ro-
bustness. RDNAS introduces a dual-branch cell with separate "normal" and "ro-
bust" pathways, fused via a lightweight attention module to capture complemen-
tary representations without significantly enlarging the search space. To reliably
score candidate operations under adversarial training, we develop **ROSE** (Robust
Outlier-Aware Shapley Estimator), which stabilizes Shapley-based evaluation via
median-of-means smoothing and interquartile-range filtering, reducing bias from
noisy gradients. RDNAS consistently discovers architectures that outperform both
hand-crafted networks and state-of-the-art robust NAS baselines across CIFAR-
10, CIFAR-100, SVHN, and Tiny-ImageNet. Notably, it achieves 52.6% $PGD^{20}$
robustness on CIFAR-10 while maintaining strong clean accuracy. Extensive abla-
tions validate the effectiveness of the dual-branch design, attention-based fusion,
and robustness-aware search. Overall, RDNAS provides a scalable and effective
framework for discovering architectures resilient to adversarial attacks.

## 1 INTRODUCTION

Deep neural networks (DNNs) have achieved strong performance across vision tasks such as classi-
fication, segmentation, and detection (Krizhevsky et al., 2012; He et al., 2015). Yet their deployment
in safety-critical domains—autonomous driving (Chen & Huang, 2017), medical imaging (Taigman
et al., 2014), or biometric authentication—remains limited by their vulnerability to adversarial per-
turbations: small, imperceptible input changes that can drastically alter predictions. This fragility
motivates research into adversarial robustness, where adversarial training (Madry et al., 2019; Zhang
et al., 2019) is the prevailing defense. While effective, adversarial training alone cannot fully exploit
robustness potential if the network architecture itself is not designed with robustness in mind.

Neural architecture search (NAS) offers a way to automatically discover architectures with strong
performance. However, most NAS approaches—including differentiable (Heuillet et al., 2024;
Yang et al., 2020; Chen et al., 2019b), zero-shot (Mellor et al., 2021; Peng et al., 2024), reinforce-
ment learning (Zoph & Le, 2017; Baker et al., 2017), or evolutionary methods (Zhou et al., 2023;
2024)—optimize almost exclusively for clean accuracy, while treating robustness as secondary. Re-
cent robust NAS studies (Simon et al., 2022; Feng et al., 2025a) incorporate adversarial training into
the search loop, but typically score candidate operations via noisy adversarial gradients, leading to
unstable updates and inconsistent robustness (Dong et al., 2025b; Cheng et al., 2023). For example,
Shapley-based methods (Xiao et al., 2022) can be skewed by outliers, yielding unreliable attribution.

We propose **RDNAS** (Robust Dual-Branch NAS), a robustness-aware framework that jointly op-
timizes clean (standard) accuracy and adversarial robustness. RDNAS builds on three ideas:
**(i)** A dual-branch cell that disentangles feature representation learning into a *normal* and a *ro-
bust* pathway, fused through a lightweight attention mechanism. Unlike ensembles or mixture-
of-experts, this design does not duplicate operators or widen the search combinatorics; in-
stead, it introduces a single extra dimension (one additional cell type) while keeping the op-
erator set unchanged, thereby maintaining search efficiency. **(ii)** Adversarial training directly
embedded in the inner loop of NAS, steering the search toward architectures that are in-

herently robust to perturbations. **(iii) ROSE** (Robust Outlier-Aware Shapley Estimator), a stabilized Shapley-based scoring mechanism that combines median-of-means smoothing with interquartile-range filtering to dampen gradient noise and reward rare but decisive operation effects.

To keep the search tractable, we couple ROSE with a few-sample evaluation strategy that reduces both the supernet training and scoring costs, achieving stability without significant computational overhead. Moreover, inspired by robustness studies on wide architectures (Wu et al., 2025), RDNAS prioritizes width over depth, achieving resilience with fewer layers and similar parameter budgets to conventional NAS cells.

We validate RDNAS on CIFAR-10 (Krizhevsky, 2009), CIFAR-100 (Krizhevsky, 2009), SVHN, and Tiny-ImageNet (Deng et al., 2009) against diverse adversarial attacks (including FGSM (Goodfellow et al., 2015), PGD (Madry et al., 2019), and transfer-based attacks (Papernot et al., 2017)). As shown in Figure 1, RDNAS consistently surpasses both hand-crafted and NAS baselines in clean and robust accuracy. Our contributions are threefold: **(i)** A dual-branch cell design that explicitly separates and fuses clean and robust pathways, providing complementary representations without inflating the search space. **(ii)** A robustness-aware scoring mechanism (ROSE) that stabilizes Shapley attribution under adversarial training through principled statistical techniques. **(iii)** Empirical evidence that RDNAS discovers architectures with superior robustness and accuracy across datasets and attacks, while maintaining computational efficiency.

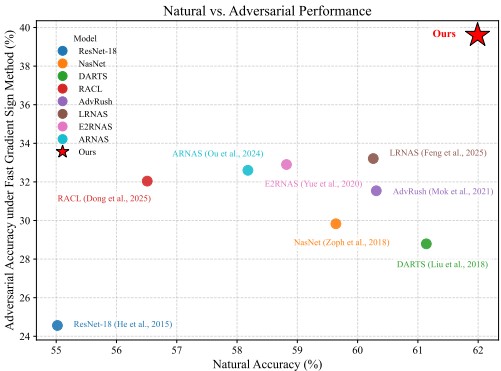

Figure 1: Clean vs. adversarial (FGSM) accuracy of various architectures on CIFAR-100. **Ours** achieves a favorable trade-off between robustness and clean performance, outperforming NAS-based and hand-crafted models.

To ensure both rigor and reproducibility, we complement our empirical study with **theoretical analysis** of the proposed estimator and **release full source code** and search logs in the appendix and supplementary material.

## 2 RELATED WORK

### 2.1 ADVERSARIAL ROBUSTNESS (ATTACKS VS. DEFENSES)

Deep neural networks (DNNs) have achieved remarkable performance across many tasks yet remain vulnerable to adversarial examples, i.e., small (often human-imperceptible) perturbations that can significantly alter model predictions. Adversarial attacks are commonly categorized into *white-box* and *black-box* settings. In the white-box case, the attacker assumes full access to model parameters and gradients: FGSM (Goodfellow et al., 2015) performs a fast single-step perturbation aligned with the loss gradient, whereas PGD (Madry et al., 2019) performs multi-step projected gradient ascent within an $\ell_\infty$ ball of radius $\epsilon$ and serves as a strong baseline for white-box evaluation. In the black-box case, attacks operate under limited information, relying either on transferability (adversarial examples crafted on a surrogate model) or on query-based methods that estimate gradients from model outputs. AutoAttack (Croce & Hein, 2020) provides a parameter-free evaluation suite that combines strong white-box attacks (APGD-CE/APGD-DLR, FAB) with a black-box component (Square), and is widely used as a reliable robustness benchmark.

To counter adversarial threats, a variety of defenses have been proposed. Among existing strategies, adversarial training has proven to be one of the most effective—in particular, PGD-based adversarial training, which is formulated as a min–max optimization between the model and the adversary. Notable extensions include TRADES (Zhang et al., 2019), which balances standard accuracy and robustness through a regularized loss. Despite these advances, the robustness of a model remains closely tied to its underlying architecture. Architectures not explicitly designed for robustness often exhibit an inherent trade-off between clean accuracy and adversarial performance. This observation underscores the need for robustness-aware NAS, where adversarial objectives are integrated directly into the search process.

## 2.2 Robustness-Aware Neural Architecture Search

Neural architecture search (NAS) automates the design of DNNs, optimizing trade-offs among accuracy, efficiency, and model complexity. Early efforts considered *macro-level* search over entire networks (Zoph & Le, 2017), followed by *micro-level* approaches that learn reusable cells and stack them to build full models (Pham et al., 2018). To improve scalability, DARTS (Liu et al., 2019) introduced a continuous relaxation that forms a Supernet with differentiable operation selection; subsequent variants such as GDAS (Dong & Yang, 2019b) and PC-DARTS (Xu et al., 2019) further improved the efficiency of gradient-based search.

While NAS has achieved strong results in clean settings, adversarial robustness has only recently become a central focus. Early work incorporated robustness into DARTS-style frameworks via adversarial training or robustness-aware objectives—e.g., smoothness priors, certified bounds, or Jacobian regularization—to guide the search (Guo et al., 2020a; Zela et al., 2020; Mok et al., 2021; Hosseini et al., 2021). Others explored broader search spaces, hybrid designs, and topological changes to improve robustness, revealing that simple strategies such as ensembling or width expansion can enhance adversarial accuracy (Vargas et al., 2019; Yang et al., 2024b; Devaguptapu et al., 2021; Liu et al., 2023). More recent methods adopt multi-objective formulations (Geraeinejad et al., 2021) to jointly consider accuracy, robustness, and efficiency, while designing specialized search spaces (Sun et al., 2025) and robustness-aware proxies (Jung et al., 2023). Additional strategies—including knowledge distillation (Nath et al., 2024), architectural disentanglement (Yang et al., 2024a), and adaptive channel allocation (Li et al., 2024)—further refine the search process (Chen et al., 2020). Meanwhile, techniques like Fair-DARTS (Chu et al., 2020) and BatchQuant (Bai et al., 2021) study robustness under constraints such as discretization and quantization.

Collectively, these developments reflect growing momentum toward NAS (especially DARTS-based) frameworks that explicitly optimize adversarial robustness across architectures and domains. Nevertheless, many approaches still depend on predefined templates and heuristic evaluations (e.g., PGD accuracy), which can introduce statistical noise and obscure robustness-critical factors. This motivates the need for principled, robustness-aware NAS frameworks that dynamically identify sensitivity in operations and architectural choices, enabling the discovery of inherently robust models.

## 2.3 Shapley Values in Architecture Search

The Shapley value, rooted in cooperative game theory (Nowak & Radzik, 1994), provides a principled way to assess the contribution of individual components to a collective reward. Let $N = \{1, \ldots, n\}$ index the components and let $v : 2^N \to \mathbb{R}$ assign a value to each subset $S \subseteq N$. The Shapley value of component $i \in N$ is

$$\phi_i(v) = \frac{1}{n!} \sum_{\pi \in \mathcal{P}(N)} \left[ v(\mathrm{pre}_i(\pi) \cup \{i\}) - v(\mathrm{pre}_i(\pi)) \right], \tag{1}$$

where $\mathcal{P}(N)$ is the set of all permutations of $N$ and $\mathrm{pre}_i(\pi)$ denotes the set of components preceding $i$ in permutation $\pi$. Shapley-based approaches have been widely used for feature attribution and, more recently, for NAS (Xiao et al., 2022) to identify influential operations. AutoShape (Fang et al., 2023) and GraphNAP++ (Oloulade et al., 2025) apply Shapley-guided pruning or architecture selection. KernelSHAP-NAS (Tran et al., 2025) adapts KernelSHAP to approximate operation influence in a supernet more robustly than earlier sampling-based methods. D-DARTS (Heuillet et al., 2023) introduces a Shapley-based loss to guide distributed differentiable NAS, promoting diversity and cell-specific optimization. Beyond architecture search, SDL (Djenouri et al., 2023) leverages Shapley values for model consensus in general-purpose vision systems, and various algorithms have been proposed to estimate Shapley values more accurately and efficiently (Chen et al., 2023).

Nevertheless, effectiveness under adversarial evaluation can be hindered by high variance in marginal contributions due to input perturbations and limited sample sizes. To address this, we develop **ROSE** with two robust statistical techniques: the interquartile range (IQR) (Rousseeuw & Hubert, 2017), which flags outlier contributions using quartiles $Q_1$ and $Q_3$ (with IQR $= Q_3 - Q_1$), and the median-of-means (MoM) estimator (Lecué & Lerasle, 2017). These tools yield a more stable and robust estimation pipeline for operation importance under adversarial noise. This design grounds ROSE in established statistical principles while extending Shapley-based NAS to adversarial settings for the first time; we further provide detailed theoretical analysis in the appendix.

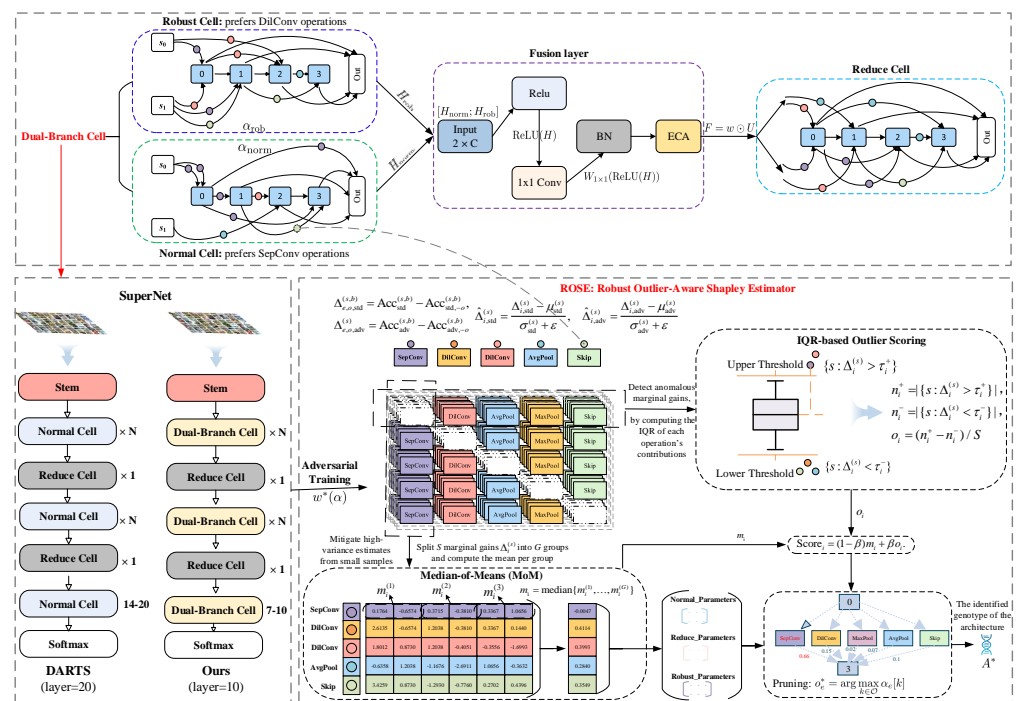

Figure 2: Overview of the RDNAS framework: (left) the dual-branch cell design; (middle) a comparison between standard DARTS (20 layers) and our dual-branch network (10 layers); (right) the ROSE estimator.

# 3 METHODOLOGY

## 3.1 OVERVIEW OF RDNAS

RDNAS follows the DARTS paradigm in which an architecture is built by stacking cells—directed acyclic graphs (DAGs) with $N$ intermediate nodes and a predefined operation set $\mathcal{O}$ (e.g., depthwise convolutions, pooling, identity). For each edge $(i,j)$, we maintain logits $\boldsymbol{\alpha}^{(i,j)} = \{\alpha_o^{(i,j)}\}_{o\in\mathcal{O}}$ that parameterize a softmax mixture of operations:

$$\tilde{o}^{(i,j)}\big(x^{(i)}\big) = \sum_{o\in\mathcal{O}} \frac{\exp(\alpha_o^{(i,j)})}{\sum_{o'\in\mathcal{O}}\exp(\alpha_{o'}^{(i,j)})}\, o\big(x^{(i)}\big). \tag{2}$$

As illustrated in Figure 2, we enhance the standard DARTS backbone—typically 20 normal cells and 2 reduction cells—by replacing each normal cell with a *dual-branch* cell while halving the depth (10 dual-branch cells plus 2 reduction cells). The two branches process features in parallel to emphasize clean and robust pathways, respectively; at each fusion point an *Efficient Channel Attention* (ECA) module adaptively reweights and aggregates channel-wise responses, enabling selective retention of informative representations (see Appendix A.2 for Grad-CAM visualizations). The final representation is fed to a softmax classifier.

To support efficient search, we relax discrete operation choices into continuous architecture parameters $\alpha$ and jointly optimize network weights $\omega$ and $\alpha$ via bilevel optimization:

$$\min_{\alpha}\ \mathcal{L}_{\text{val}}\big(\omega^*(\alpha), \alpha\big), \tag{3}$$

$$\text{s.t.}\ \omega^*(\alpha) = \arg\min_{\omega}\ \mathcal{L}_{\text{train}}(\omega, \alpha), \tag{4}$$

where the inner objective adopts a composite loss on clean and adversarial examples,

$$\mathcal{L}_{\text{train}} = \lambda \cdot \mathcal{L}_{\text{clean}} + (1-\lambda) \cdot \mathcal{L}_{\text{adv}}. \tag{5}$$

(details in Sec. 4.2), and the outer objective is evaluated under the same robustness settings. RDNAS employs robustness-aware bilevel optimization to balance accuracy and adversarial resilience.

To improve stability under noisy gradients, we introduce **ROSE**, which scores operations on each edge $(i, j)$ and guides lightweight pruning. By combining MoM smoothing with IQR filtering, ROSE yields robust importance estimates. After convergence, we discretize by retaining the top operation(s) per edge based on ROSE scores, producing a compact model that balances clean accuracy and adversarial robustness.

## 3.2 Dual-Branch Cell Design

Conventional gradient-based NAS methods (e.g., DARTS) rely on a *single* shared cell to process both clean and adversarial data. Under adversarial training, this shared cell must simultaneously serve two competing objectives, and we empirically observe that the resulting architectures tend to favor either clean accuracy or robustness, but rarely both. Although one could attempt a single cell with a composite loss (e.g., TRADES (Zhang et al., 2019)), we find that explicitly separating the optimization paths allows each branch to specialize, leading to superior overall performance. This design reduces interference between objectives and provides more stable architecture search under adversarial training.

To this end, we introduce the *Dual-Branch Cell*, which decouples learning into two parallel sub-networks: a normal branch for clean accuracy and a robust branch for adversarial resilience. Both branches process the same inputs $(s_0, s_1)$ using a shared operator class $\mathcal{C}$ but independent architecture parameters $\alpha_{\text{norm}}$ and $\alpha_{\text{rob}}$:

$$H_{\text{norm}} = \mathcal{C}(s_0, s_1; \alpha_{\text{norm}}), \quad H_{\text{rob}} = \mathcal{C}(s_0, s_1; \alpha_{\text{rob}}), \tag{6}$$

where $H_{\text{norm}}, H_{\text{rob}} \in \mathbb{R}^{C \times H \times W}$. Their outputs are concatenated along the channel dimension to form a joint representation:

$$H = [H_{\text{norm}}; H_{\text{rob}}] \in \mathbb{R}^{2C \times H \times W}. \tag{7}$$

Naively fusing both branches without weighting treats clean and robust features as equally important at all times, which can limit expressiveness. To more effectively exploit the complementary representations, we adopt an attention-based fusion mechanism inspired by Efficient Channel Attention (ECA) (Wang et al., 2020). First, we reduce the channel dimension with a $1 \times 1$ convolution followed by ReLU activation:

$$U = W_{1 \times 1}(\text{ReLU}(H)) \in \mathbb{R}^{C \times H \times W}. \tag{8}$$

Next, global average pooling extracts a compact channel descriptor:

$$z = \text{GAP}(U) \in \mathbb{R}^C, \tag{9}$$

which is passed through a 1D convolution (with an adaptively determined kernel size as in ECA) and a sigmoid activation to generate channel-wise attention weights:

$$w = \sigma(\text{Conv1D}(z)) \in (0, 1)^C. \tag{10}$$

These weights modulate the feature map via element-wise multiplication:

$$F = w \odot U \in \mathbb{R}^{C \times H \times W}. \tag{11}$$

The fused output $F$ is forwarded to the next layer. By explicitly separating optimization paths and adaptively fusing them, the Dual-Branch Cell encourages each branch to specialize: the normal branch can focus on fine-grained discriminative patterns, while the robust branch learns smoother, perturbation-insensitive structures. This reduces conflict between clean and robust objectives and stabilizes architecture search. Our ablation in Table 4 shows that introducing ECA-based fusion consistently improves both clean and PGD$^{20}$ accuracy over a dual-branch design without attention. Moreover, Grad-CAM visualizations in Appendix A.2 qualitatively confirm that the two branches attend to different yet complementary regions under clean and adversarial inputs, providing empirical evidence that the dual-branch design effectively captures diverse features.

## 3.3 Adversarially Robust Search

Following recent advances in robust NAS, we integrate adversarial training directly into the inner optimization loop to promote robustness-aware search. While conventional DARTS assumes subnet

---

**Algorithm 1** Adversarially Robust Architecture Search

---

1: **Input:** Search space $\mathcal{S} = \{o_{ij}\}$, warm-up epochs $N_w$, search epochs $N_s$, #Shapley samples $S$, #MoM groups $G$, IQR sensitivity $\gamma$, training data $D_{\text{train}}$, validation data $D_{\text{val}}$
2: **Output:** Adversarially robust architecture $A^*$
3: Initialize architecture logits $\alpha_n \in \mathbb{R}^{E \times O}$ for each branch $b \in \{\text{normal}, \text{reduce}, \text{robust}\}$ and weights $\omega$;
4: **for** epoch = 1 **to** $N_w + N_s$ **do**
5:     Train $\omega$ on $D_{\text{train}}$ by minimize the composite training loss in (5);
6:     **if** epoch $> N_w$ **then**
7:         Compute the clean and adversarial accuracy deltas $\Delta_{e,o,\text{std}}^{(s,b)}$ and $\Delta_{e,o,\text{adv}}^{(s,b)}$ by (15);
8:         Compute the final $\text{Score}_{e,o}^{(b)}$ by ROSE in (19) based on (15) to (18);
9:         Update logits: $\alpha \leftarrow \alpha + \eta \cdot \text{Score}$;
10:        Row-normalize each edge of $\alpha$ so its logits sum to 1;
11:     **end if**
12: **end for**
13: Pick the operation for each edge with highest logit to form $A^*$;
14: **Return:** $A^*$

---

ranking is consistent between small-search and large-inference networks, this assumption often fails under adversarial training due to high-variance gradients. To address this, we leverage ROSE as a statistically stable operation scorer, which mitigates the effect of noisy gradients and provides reliable guidance for architecture updates.

Specifically, during each epoch, clean inputs $x$ are transformed into adversarial examples $x^{\text{adv}}$ using a fast PGD-$k$ method (e.g., $k = 7$ steps):

$$x^{\text{adv}} = \arg \max_{\|\delta\|_\infty \leq \epsilon} \mathcal{L}_{\text{train}}(f(x + \delta), y), \tag{12}$$

where $\epsilon$ constrains the perturbation magnitude and $\mathcal{L}_{\text{train}}$ is the composite training loss that mitigates overfitting to clean samples and steers weight learning toward robust representations.

After obtaining the optimal network weights $w^*(\alpha)$, the outer optimization fixes $w^*(\alpha)$ and updates architecture parameters $\alpha$ by minimizing the ROSE-weighted validation loss:

$$\mathcal{L}_{\text{val}}(w^*(\alpha), \alpha) \;=\; -\sum_{b=1}^{3} \sum_{e=1}^{E} \sum_{o=1}^{O} p_{e,o}^{(b)}(\alpha) \, \text{Score}_{e,o}^{(b)} \tag{13}$$

where $b$ indexes normal, reduce, and robust branches, and $p_{e,o}^{(b)}(\alpha) = \text{softmax}(\alpha_{e,:}^{(b)})_o$ is the probability of selecting operation $o$ on edge $e$ of branch $b$. The ROSE score $\text{Score}_{e,o}^{(b)}$ stabilizes operation evaluation under noisy adversarial gradients, ensuring that the outer-loop update favors consistently robust operations.

Upon convergence, architecture discretization selects the most probable operation per edge:

$$o_e^* = \arg \max_{k \in \mathcal{O}} \alpha_e[k] \tag{14}$$

The resulting normal and robust cells are paired into Dual-Branch Cells, interleaved with Reduce Cells, and stacked to form a compact backbone that balances clean accuracy and adversarial robustness. To reduce search cost and maintain practical efficiency, we adopt a small-sample search strategy: only a fraction of the training data is used per search epoch, which, combined with lightweight 10-cell networks, drastically reduces computation while preserving ranking stability under ROSE guidance. The overall search procedure is summarized in Algorithm 1. This setup ensures that RDNAS discovers robust architectures efficiently, mitigating both the DARTS-related ranking inconsistencies and the high computational overhead of adversarial training.

### 3.4 ROSE: ROBUST OUTLIER-AWARE SHAPLEY ESTIMATOR

After revisiting the Shapley foundations, we find that certain operations disproportionately influence adversarial robustness. However, under adversarial training, architecture evaluation is inherently noisy: gradient estimates fluctuate due to stochastic perturbations, and individual operations may

appear important or unimportant depending on the sampled adversarial examples. This motivates ROSE, which refines Shapley-based scoring to capture both stable contributions and rare but critical effects. Unlike conventional approaches that score all operations jointly, ROSE applies separate evaluations to the normal, reduction, and robust cells, yielding three branch-specific score matrices. By decoupling branches, ROSE also preserves branch-specific sensitivity, aligning with our dual-branch cell design.

ROSE enhances the standard Monte Carlo Shapley sampler using two complementary techniques: 1) *IQR-based outlier detection*, operations whose standardized marginal gains fall outside the interquartile range are highlighted, allowing rare yet decisive contributions to be recognized rather than averaged away. This helps capture edges that are critical under adversarial conditions. 2) *MoM smoothing*, by partitioning permutations into groups and aggregating their means, MoM reduces the variance introduced by stochastic gradients, producing a stable Shapley estimate that reliably ranks operations across noisy adversarial evaluations.

Let $b \in \{\text{normal}, \text{reduce}, \text{robust}\}$ index each cell branch, and $\mathcal{O}^{(b)}$ denote its candidate operations, with the rest of the supernet fixed. Over $S$ random permutations, we compute clean and adversarial accuracy deltas for each operation $o$ on edge $e$ in branch $b$:

$$\Delta_{e,o,\text{std}}^{(s,b)} = \text{Acc}_{\text{std}}^{(s,b)} - \text{Acc}_{\text{std},-o}^{(s,b)}, \quad \Delta_{e,o,\text{adv}}^{(s,b)} = \text{Acc}_{\text{adv}}^{(s,b)} - \text{Acc}_{\text{adv},-o}^{(s,b)} \tag{15}$$

where $\text{Acc}_{\text{std}}^{(s,b)}$ (or $\text{Acc}_{\text{adv}}^{(s,b)}$) denotes validation accuracy with operation $o$, and $\text{Acc}_{\text{std},-o}^{(s,b)}$ (or $\text{Acc}_{\text{adv},-o}^{(s,b)}$) denotes the accuracy after removing $o$. These deltas are standardized per permutation to reduce variance:

$$\hat{\Delta}_{i,\text{std}}^{(s)} = \frac{\Delta_{i,\text{std}}^{(s)} - \mu_{\text{std}}^{(s)}}{\sigma_{\text{std}}^{(s)} + \epsilon}, \quad \hat{\Delta}_{i,\text{adv}}^{(s)} = \frac{\Delta_{i,\text{adv}}^{(s)} - \mu_{\text{adv}}^{(s)}}{\sigma_{\text{adv}}^{(s)} + \epsilon} \tag{16}$$

where $\mu^{(s)}$ and $\sigma^{(s)}$ are empirical statistics across operations, and $\epsilon$ ensures numerical stability.

To capture anomalies, we compute the IQR for each $\{\hat{\Delta}_{e,o}^{(s,b)}\}_{s=1}^{S}$, set upper and lower thresholds $\tau_{e,o}^{(b)+}$ and $\tau_{e,o}^{(b)-}$, and derive an outlier score:

$$v_{e,o}^{(b)} = \frac{\left| \{s : \hat{\Delta}_{e,o}^{(s,b)} > \tau_{e,o}^{(b)+}\} \right| - \left| \{s : \hat{\Delta}_{e,o}^{(s,b)} < \tau_{e,o}^{(b)-}\} \right|}{S} \tag{17}$$

Here, $v_{e,o}^{(b)} > 0$ suggests frequent robustness gains when $o$ is dropped, while $v_{e,o}^{(b)} < 0$ indicates critical importance.

For stability, the MoM estimate is computed by partitioning samples into $G$ folds, averaging within each fold, and taking the median:

$$m_{e,o}^{(b)} = \text{median}\left\{ \frac{1}{|\mathcal{G}_k|} \sum_{s \in \mathcal{G}_k} \hat{\Delta}_{e,o}^{(s,b)} \right\}_{k=1}^{G} \tag{18}$$

Intuitively, MoM ensures that the ranking of operations is not dominated by extreme but noisy gradients, while IQR ensures that rare but impactful operations are not ignored. This combination provides a principled, statistically robust evaluation of each candidate.

The final ROSE score combines both components:

$$\text{Score}_{e,o}^{(b)} = (1 - \beta)\, m_{e,o}^{(b)} + \beta\, v_{e,o}^{(b)} \tag{19}$$

where $\beta \in [0, 1]$ balances *steady reward* and *occasional indispensability*. Empirically, $\beta \in [0.3, 0.5]$ yields architectures that are both reliably accurate and resilient under adversarial conditions, demonstrating that ROSE effectively guides the search toward robust-optimal configurations.

We use the ROSE score to rank (and, if enabled, lightly prune) candidate operations in each cell. This selective mechanism stabilizes operation evaluation under noisy adversarial training and highlights components that are consistently valuable or occasionally critical, providing a clear justification for ROSE's necessity. Sensitivity experiments on $\beta$ are presented in Sec. A.3, with additional hyperparameter studies for the entire framework provided in Appendix A.3.

# 4 EXPERIMENTAL STUDIES

## 4.1 BENCHMARK DATASETS AND BASELINES

Following established practices in robust NAS research (Guo et al., 2019; Mok et al., 2021), we evaluate our proposed method, RDNAS, on four benchmark datasets: CIFAR-10, CIFAR-100 (Krizhevsky, 2009), SVHN, and Tiny-ImageNet-200 (Le & Yang, 2015).

We compare RDNAS against a comprehensive set of peer methods, including manually designed architectures such as ResNet-18 (He et al., 2015) and DenseNet-121 (Huang et al., 2017), standard NAS frameworks like DARTS (Liu et al., 2019), NASNet (Zoph et al., 2018), and PDARTS (Chen et al., 2019a), GDAS (Dong & Yang, 2019b), SETN (Dong & Yang, 2019a), and ENAS (Pham et al., 2018); training-free NAS baselines including TTNAS (Lin et al., 2024) and MOTE-NAS (Zhang et al., 2024); as well as robust NAS approaches such as RobNet (Guo et al., 2020b), AdvRush (Mok et al., 2021), RACL (Dong et al., 2025a), E2RNAS (Yue et al., 2020), ARNAS (Ou et al., 2024), and LRNAS (Feng et al., 2025a). We ensure fair comparisons by reporting FLOPs and parameter counts for all baselines, highlighting that improvements of RDNAS are achieved under comparable or lower computational budgets.

## 4.2 SEARCH CONFIGURATION AND EVALUATION PROTOCOL

Given that RDNAS adopts a shallower architecture, we conduct the search on a 10-cell network (instead of 8) and *adjust* the initial channel count to 32 (the common setting is 36) to maintain training–search consistency. The search runs for 50 epochs using adversarial training based on 7-step PGD with a step size of $2/255$ and perturbation bound of $8/255$. The adversarial loss regularization coefficient is set to 0.5. Other settings follow DARTS. Network weights ($\omega$) are optimized using SGD with momentum 0.9, learning rate $\eta_w = 0.025$ (cosine decay), and weight decay $3 \times 10^{-4}$. Architecture parameters ($\alpha$) use Adam with $\eta_\alpha = 3 \times 10^{-4}$, $\beta = (0.5, 0.999)$, and weight decay $10^{-3}$. To substantially reduce search cost and accelerate convergence under adversarial training, we adopt a small-sample search: on CIFAR-10, only 1,000 training samples and 500 validation samples are used. **ROSE is critical here**, as it stabilizes architecture ranking under small-sample noisy gradients, enabling reliable discovery of robust architectures in a fraction of typical search time. To the best of our knowledge, this is the first attempt to combine small-sample one-shot NAS with adversarial training.

For final evaluation, we stack 10 cells with an initial channel count of 32. Stem / Reduction-1 / Reduction-2 channels are set to 32 / 128 / 128. Following AdvRush (Mok et al., 2021), we adversarially train each model for 120 epochs using 7-step PGD (step size 0.01, $\epsilon = 8/255$). Optimization uses SGD with momentum 0.9 and weight decay $1 \times 10^{-4}$. Learning rates are 0.1 (CIFAR-10/100) and 0.01 (SVHN), decayed at epochs 90 and 104. Batch size is 32. All experiments are performed on an NVIDIA GeForce RTX 4080 Super GPU.

Table 1: Robust accuracy on CIFAR-10. All attacks use $\ell_\infty$ budget $\epsilon = 8/255$. Columns report clean (Nat.) and white-box attacks; the last two columns show AutoAttack (AA) and GPU-days. Best in each column is **bold**.

| Model | Params | FLOPs | Evaluation ($\ell_\infty$) | | | | | AA | GPUday |
|---|---|---|---|---|---|---|---|---|---|
| | | | Nat. | FGSM | PGD$^{20}$ | PGD$^{100}$ | APGD-CE | | |
| ResNet-18 | 11.2M | 37.67M | 84.09% | 54.64% | 45.86% | 45.53% | 44.54% | 43.22% | – |
| DenseNet-121 | 7.0M | 59.83M | 85.95% | 58.46% | 50.49% | 49.92% | 49.11% | 47.46% | – |
| DARTS | 3.3M | 547.44M | 85.92% | 58.96% | 51.45% | 49.32% | 48.32% | 47.73% | 1.0 |
| PDARTS | 3.4M | 550.75M | 85.38% | 59.12% | 51.32% | 50.91% | 49.96% | 48.52% | 0.3 |
| RobNet | 5.6M | 800.40M | 85.00% | 59.22% | 52.09% | 51.14% | 50.41% | 48.56% | 3.3 |
| DSRNA | 2.0M | 336.23M | 80.93% | 54.49% | 49.11% | 48.89% | 48.67% | 44.87% | 0.4 |
| LRNAS | 2.2M | 346.10M | 84.26% | 59.89% | 50.20% | 49.90% | 49.83% | 49.07% | 0.4 |
| RACL | 3.6M | 568.86M | 85.13% | 59.45% | 51.89% | 51.63% | 51.09% | **50.23%** | 0.5 |
| AdvRush | 4.2M | 668.53M | 86.38% | 60.32% | 52.29% | 51.80% | 51.42% | 50.05% | 0.7 |
| **RDNAS (Ours)** | 4.4M | 1.30G | **86.56%** | **60.44%** | **52.62%** | **52.24%** | **52.05%** | 49.98% | 0.2 |

Table 2: Transfer-based black-box accuracy on CIFAR-10/100 (%).

| Dataset | Params (M) | Source | Target | | | | | |
|---|---|---|---|---|---|---|---|---|
| | | | DARTS | AdvRush | LRNAS | RACL | PDARTS | Ours |
| CIFAR-10 | 3.3 | DARTS | – | 65.22 | 63.01 | 65.22 | 66.21 | **67.54** |
| | 4.2 | AdvRush | 65.40 | – | 63.52 | 65.45 | 66.29 | **67.60** |
| | 2.3 | LRNAS | 67.96 | 68.67 | – | 67.83 | 68.92 | **69.46** |
| | 3.6 | RACL | 65.44 | 65.46 | 62.54 | – | 64.97 | **65.80** |
| | 3.4 | PDARTS | 65.25 | 65.24 | 62.61 | 63.81 | – | **65.52** |
| | 4.4 | Ours | **67.16** | 66.80 | 63.85 | 65.79 | 66.32 | – |
| CIFAR-100 | 3.3 | DARTS | – | 43.63 | 42.78 | 43.29 | 40.17 | **46.37** |
| | 4.2 | AdvRush | 42.46 | – | 41.91 | 42.83 | 40.01 | **46.54** |
| | 2.8 | LRNAS | 43.35 | 44.27 | – | 44.73 | 41.87 | **47.06** |
| | 3.6 | RACL | 42.68 | 41.83 | 42.25 | – | 40.39 | **46.72** |
| | 3.4 | PDARTS | 40.06 | 41.39 | 38.38 | 40.71 | – | **45.94** |
| | 4.4 | Ours | 47.04 | **47.83** | 44.61 | 47.65 | 47.18 | – |

Table 3: Cross-dataset transfer results (%, $\epsilon=8/255$, PGD$^{20}$ step size $2/255$). Best result in each sub-table is **bold**.

| **CIFAR-100** | | | |
|---|---|---|---|
| Model | Clean | FGSM | PGD$^{20}$ |
| ResNet-18 | 55.57 | 26.03 | 21.44 |
| AdvRush | 60.31 | 31.54 | 27.38 |
| RACL | 59.18 | 34.40 | 30.41 |
| ARNAS | 58.18 | 32.60 | 29.54 |
| **RDNAS** | **61.99** | **39.60** | 29.42 |

| **SVHN** | | | |
|---|---|---|---|
| Model | Clean | FGSM | PGD$^{20}$ |
| ResNet-18 | 92.06 | 88.73 | 69.51 |
| DenseNet-121 | 95.10 | 93.01 | 89.58 |
| ARNAS | 95.84 | 94.43 | 92.02 |
| AdvRush | 96.53 | 94.95 | 91.14 |
| **RDNAS** | **97.88** | **96.37** | **95.80** |

| **Tiny-ImageNet-200** | | | |
|---|---|---|---|
| Model | Clean | FGSM | PGD$^{20}$ |
| ResNet-18 | 36.26 | 16.08 | 13.94 |
| DenseNet-121 | 47.56 | 22.98 | 18.06 |
| PDARTS | 45.94 | 24.36 | 22.74 |
| AdvRush | 46.42 | 24.20 | **22.89** |
| **RDNAS** | **56.84** | **24.49** | 19.40 |

## 4.3 RESULTS ON WHITE-BOX ATTACKS

Table 1 shows that **RDNAS** attains the strongest overall white-box results: it is best on Clean, FGSM, PGD$^{20}$, PGD$^{100}$, and APGD-CE, and only $0.25$ pp behind the best prior AutoAttack score (49.98% vs. 50.23% for RACL), with a moderate 4.4M parameters. Our FLOPs (1.30G) are higher than some DARTS-style baselines due to the dual-branch design with ECA fusion, but this extra compute yields consistent gains across all white-box attacks. The cost is practical: the search budget is small (0.2 GPU-days), and the halved depth (10 cells) helps latency despite per-cell compute. Moreover, the robustness *transfers*: Table 2 shows best or tied-best transfer-based black-box performance in most source→target pairs, and Table 3 confirms strong cross-dataset results (CIFAR-100, SVHN, Tiny-ImageNet-200). Overall, the modest FLOPs increase buys attack- and dataset-general robustness, making the accuracy–compute trade-off favorable.

## 4.4 RESULTS ON BLACK-BOX ATTACKS

We evaluate transfer-based black-box robustness on CIFAR-10 and CIFAR-100 using adversarial examples generated by PGD (untargeted, $\epsilon = 8/255$, step size $2/255$, 20 steps, random starts) from five source models: DARTS, PDARTS, RACL, LRNAS, and AdvRush. Unless otherwise noted, attacks are single-source transfers (no ensemble). As summarized in Table 2, RDNAS outperforms peers in most transfer settings. Notably on CIFAR-100, it retains top robustness when attacked by adversarial inputs generated from the above sources. Despite its moderate size (4.4M parameters), RDNAS achieves 65.5%–69.5% robust accuracy on CIFAR-10 and 45.9%–47.8% on CIFAR-100, offering favorable robustness–parameter trade-offs relative to smaller models.

## 4.5 CROSS-DATASET TRANSFERABILITY

To test generalization, we transfer the RDNAS architecture searched on CIFAR-10 to CIFAR-100, SVHN, and Tiny-ImageNet-200, and retrain from scratch on each target dataset (no re-search). As shown in Table 3, RDNAS attains strong clean and adversarial accuracies. On CIFAR-100, it surpasses ResNet-18 and NASNet in robustness; on SVHN, it achieves 97.88% clean accuracy and 96.37% FGSM robustness; on Tiny-ImageNet-200, it delivers 56.84% clean accuracy and remains competitive under PGD, trailing AdvRush slightly.

Table 4: Ablation of ECA, adversarial-training during search, and search space. Metrics are Top-1 (%) on clean and **PGD**$^{20}$ ($\epsilon = 8/255$, step size $2/255$); Params in millions.

| ID | ECA | Adv. search | Search space | Params (M) | Clean (%) | PGD$^{20}$ (%) |
|---|---|---|---|---|---|---|
| A | × | × | Single Cell (20 cells) | 4.20 | 84.8 | 50.3 |
| B | × | ✓ | Single Cell (20 cells) | 4.28 | 85.6 | 51.0 |
| C | ✓ | ✓ | Single Cell (20 cells) | 4.30 | 86.0 | 51.5 |
| D | × | ✓ | Dual-Branch Cell (10 cells) | 4.26 | 85.7 | 51.8 |
| E | ✓ | ✓ | Dual-Branch Cell (10 cells) | 4.32 | **86.5** | **52.6** |

## 4.6 ABLATION STUDIES

**Search-Space Ablation:** As shown in Table 4, a reduced RDNAS search space (10 operations) outperforms the full DARTS space (20 operations) under FGSM and PGD, while maintaining similar clean accuracy. This demonstrates that our carefully designed operation set suffices for robust architecture discovery and reduces search complexity.

**Adversarial Training Ablation:** Clean-only search yields strong natural accuracy but poor PGD robustness. Integrating adversarial training during search significantly improves robustness, validating its importance.

**Attention Module Ablation:** Removing the Efficient Channel Attention (ECA) module slightly degrades clean accuracy and noticeably reduces adversarial robustness. Reintroducing ECA improves both, confirming its effectiveness. Grad-CAM visualizations (Appendix A.2) further show that the two branches focus on complementary regions, providing qualitative support for the attention module's role in robust feature integration.

**ROSE Ablation:** ROSE is enabled by default in our adversarial-search setting. To assess its contribution, we run an additional search variant without ROSE, using standard Shapley estimates. This variant shows higher run-to-run variability and slightly lower PGD robustness, indicating that ROSE stabilizes small-sample adversarial NAS and aids in discovering architectures with more consistent robust performance.

## 4.7 RESULTS ON NAS-BENCH-201 (CIFAR-10)

Table 5 reports validation/test accuracy and search cost on NAS-Bench-201 (CIFAR-10). We list classic gradient-based baselines and recent training/evaluation-free methods (e.g., TTNAS (Lin et al., 2024), MOTE-NAS (Zhang et al., 2024)). Our method achieves the best trade-off between accuracy and search time. This experiment demonstrates that, even under a few-sample search regime, the ROSE estimator remains stable and reliably discovers robust architectures.

| Method | Search (s) | val | test |
|---|---|---|---|
| DARTS-V2 | 35781.80 | 39.77±0.00 | 54.30±0.00 |
| GDAS | 31609.80 | 89.89±0.08 | 93.61±0.09 |
| SETN | 34139.53 | 84.04±0.28 | 87.64±0.00 |
| ENAS | 14058.80 | 37.51±3.19 | 53.89±0.58 |
| TTNAS (training-free) | 1146 | 91.02±0.11 | 93.94±0.38 |
| MOTE-NAS (K=5) | 2239 | 90.89±0.13 | 93.86±0.15 |
| **Ours** | **731** | **91.13±0.36** | **93.97±0.35** |
| **Ground Truth** | — | **91.61** | **94.37** |

Table 5: Validation and test accuracy with corresponding search cost on NAS-Bench-201 (CIFAR-10). RDNAS achieves the best trade-off between accuracy and efficiency, demonstrating stable performance even under limited search budgets. The results include the average and standard deviations for 3 runs.

## 5 CONCLUSION

This work presents RDNAS, a robust NAS framework that jointly optimizes for accuracy and adversarial robustness. We propose a dual-branch cell search space and enhance Shapley-value estimation using MoM and IQR to identify critical operations. Extensive experiments across multiple benchmarks and attack settings demonstrate the effectiveness and transferability of RDNAS, with results highlighting that heterogeneous cell placement across depths is crucial for balancing robustness and accuracy.

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

## A  APPENDIX

### A.1  THEORETICAL JUSTIFICATION OF ROSE ESTIMATOR

This appendix provides a concentration bound for the Robust Outlier-aware Shapley Estimator (ROSE) under mild assumptions.

### A.1.1 ASSUMPTIONS

**Assumption 1 (Bounded Marginal Gains).** For any branch $b$, edge $e$, operation $o$, and permutation $s$,

$$\left|\Delta_{e,o,\text{std}}^{(s,b)}\right| \leq M, \qquad \left|\Delta_{e,o,\text{adv}}^{(s,b)}\right| \leq M, \tag{20}$$

where $M > 0$ is a finite constant. Since the Shapley value $\phi_{e,o}^{(b)}$ is a convex combination of bounded marginal gains, we also have $|\phi_{e,o}^{(b)}| \leq M$.

**Assumption 2 (Permutation Independence).** The $S$ Monte-Carlo permutations are sampled independently and partitioned into buckets for the MoM estimator; independence implies inter-bucket independence.

### A.1.2 TRIANGLE INEQUALITY FOR ROSE SCORE

ROSE combines a Median-of-Means (MoM) estimator $m_{e,o}^{(b)}$ and an outlier-aware score $v_{e,o}^{(b)} \in [-1,1]$:

$$\text{Score}_{e,o}^{(b)} = (1-\beta)\, m_{e,o}^{(b)} + \beta\, v_{e,o}^{(b)}, \qquad 0 \leq \beta < 1. \tag{21}$$

Subtracting the true Shapley value $\phi_{e,o}^{(b)}$ gives

$$\left|\text{Score}_{e,o}^{(b)} - \phi_{e,o}^{(b)}\right| = \left|(1-\beta)\big(m_{e,o}^{(b)} - \phi_{e,o}^{(b)}\big) + \beta\big(v_{e,o}^{(b)} - \phi_{e,o}^{(b)}\big)\right| \tag{22}$$

$$\leq (1-\beta)\left|m_{e,o}^{(b)} - \phi_{e,o}^{(b)}\right| + \beta\left|v_{e,o}^{(b)} - \phi_{e,o}^{(b)}\right|. \tag{22$'$}$$

Because $|v_{e,o}^{(b)}| \leq 1$ and $|\phi_{e,o}^{(b)}| \leq M$, define $C := 1 + M$ for later use.

### A.1.3 PROBABILITY RELAXATION

We upper-bound the deviation probability:

$$\Pr\big(|\text{Score}_{e,o}^{(b)} - \phi_{e,o}^{(b)}| \geq \varepsilon\big). \tag{23}$$

When $\varepsilon > \beta C$, inequality equation 22$'$ implies

$$|m_{e,o}^{(b)} - \phi_{e,o}^{(b)}| \geq \frac{\varepsilon - \beta C}{1 - \beta}. \tag{24}$$

Hence

$$\Pr\big(|\text{Score}_{e,o}^{(b)} - \phi_{e,o}^{(b)}| \geq \varepsilon\big) \leq \Pr\Big(|m_{e,o}^{(b)} - \phi_{e,o}^{(b)}| \geq \tfrac{\varepsilon - \beta C}{1-\beta}\Big). \tag{25}$$

### A.1.4 CONCENTRATION OF MoM ESTIMATOR

According to the MoM concentration bound (Lugosi & Mendelson, 2019), if the bucket (group) size is at least $n_0$, then for any $t > 0$,

$$\Pr\big(|m_{e,o}^{(b)} - \phi_{e,o}^{(b)}| \geq t\big) \leq 2\exp\big(-\tfrac{n_0\, t^2}{2M^2}\big). \tag{26}$$

Plugging $t = \frac{\varepsilon - \beta C}{1-\beta}$ into equation 26 and using equation 25 yields

$$\Pr\big(|\text{Score}_{e,o}^{(b)} - \phi_{e,o}^{(b)}| \geq \varepsilon\big) \leq 2\exp\Big(-\frac{n_0(\varepsilon - \beta C)^2}{2M^2(1-\beta)^2}\Big). \tag{27}$$

### A.1.5 FINAL RELAXATION

If $\varepsilon > 2\beta C$ then $(\varepsilon - \beta C)^2 \geq \varepsilon^2/4$ and $(1-\beta)^2 \leq 1$, so

$$\Pr\big(|\text{Score}_{e,o}^{(b)} - \phi_{e,o}^{(b)}| \geq \varepsilon\big) \leq 2\exp\Big(-\frac{n_0\,\varepsilon^2}{8M^2}\Big). \tag{28}$$

$$\varepsilon > 2\beta(1 + M).$$

Thus, ROSE enjoys a sub-Gaussian tail (up to a constant factor), completing the proof. Equation equation 28 establishes a sub-Gaussian tail

$$\Pr\big(|\text{Score} - \phi| \geq \varepsilon\big) \leq 2\exp\Big(-\frac{n_0\varepsilon^2}{8M^2}\Big).$$

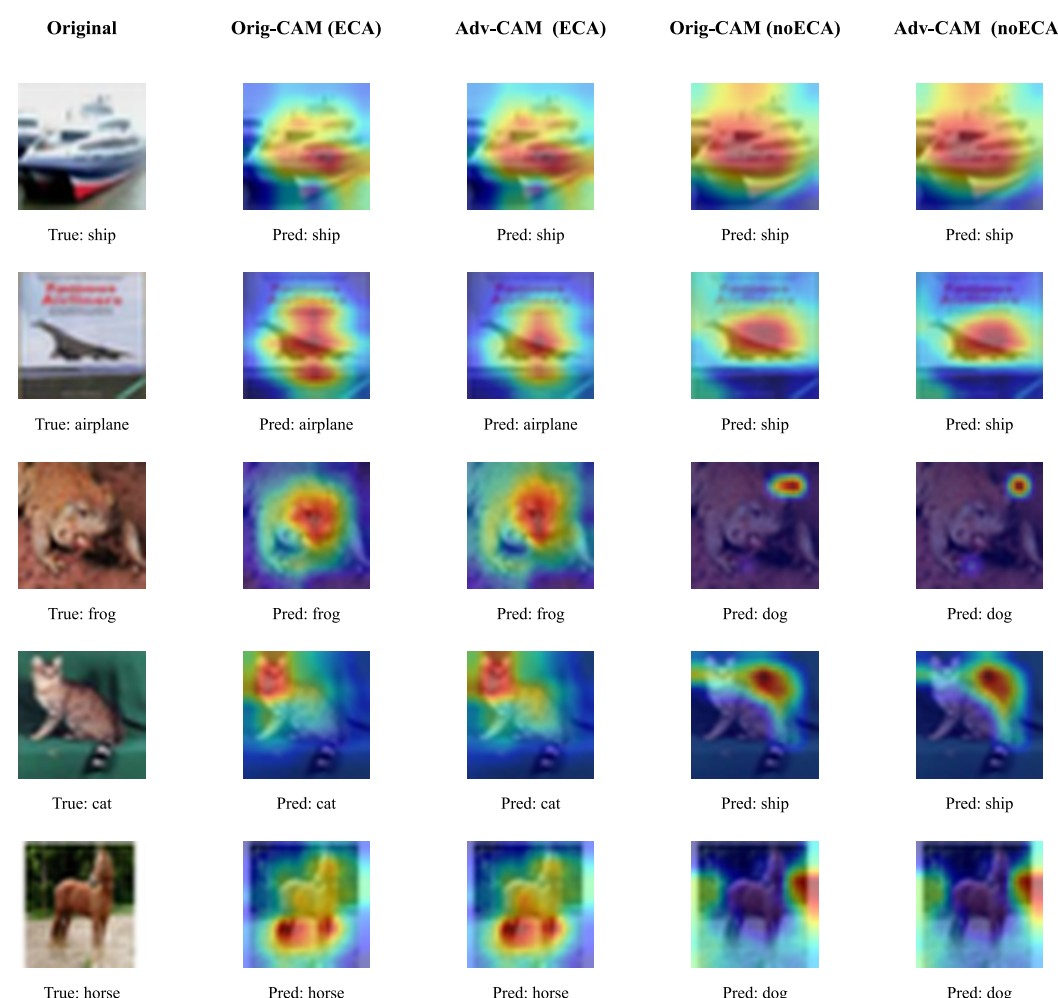

| Original | Orig-CAM (ECA) | Adv-CAM (ECA) | Orig-CAM (noECA) | Adv-CAM (noECA) |
|---|---|---|---|---|
| True: ship | Pred: ship | Pred: ship | Pred: ship | Pred: ship |
| True: airplane | Pred: airplane | Pred: airplane | Pred: ship | Pred: ship |
| True: frog | Pred: frog | Pred: frog | Pred: dog | Pred: dog |
| True: cat | Pred: cat | Pred: cat | Pred: ship | Pred: ship |
| True: horse | Pred: horse | Pred: horse | Pred: dog | Pred: dog |

Figure 3: Grad-CAM visualizations comparing models with and without ECA under clean and adversarial inputs.

**Discussion.** The MoM mechanism counteracts heavy-tailed noise, while the outlier-aware term $v_{e,o}$ suppresses extreme gradients, yielding robustness even when adversarial or extreme samples are present. The guarantees rely only on inter-bucket independence and thus extend naturally to multi-GPU or multi-node distributed settings. The error–probability relation also couples the bucket size $n_0$ with any desired confidence level, allowing practitioners to minimize sampling—and hence search cost—while meeting accuracy targets. Finally, the coefficient $\beta$ acts as a design knob between statistical reliability and outlier penalization; the theory requires $\varepsilon > 2\beta(1 + M)$, so exponentially decaying $\beta$ or rescaling gains (reducing $M$) keeps evaluations within the admissible region, transforming heuristic "$\beta$ tuning" into a quantitatively grounded procedure.Although we conduct the search with only 1k/0.5k labeled samples on CIFAR-10, our theoretical guarantees concern the *estimation* sample size used by ROSE (i.e., the number of independent evaluations per MoM bucket), rather than the dataset size itself. By increasing the number of permutations, data augmentations, and resamplings, we effectively enlarge $n_0$. With a small $\beta$ and bounded gains (small $M$), the sub-Gaussian tail bound still applies—albeit with a looser constant— hence our small-data search remains consistent with the theory.

## A.2 VISUALIZATION OF ECA MODULE'S EFFECT

To examine how the ECA (Efficient Channel Attention) module affects robustness and interpretability, we visualize class activation maps (CAMs) using Grad-CAM (Selvaraju et al., 2019) under both

clean and adversarial inputs. We use the last convolutional block for Grad-CAM, and normalize each CAM to $[0, 1]$ (per image) before overlay. Adversarial CAMs are generated with untargeted PGD-20 under $\ell_\infty$ budget $\epsilon = 8/255$ (step size $2/255$, random starts). Figure 3 shows a grid of five representative CIFAR-10 samples and their corresponding CAMs, using the same inputs across models for a fair comparison.

- **Column 1**: Original image with ground-truth label.
- **Columns 2–3 (with ECA)**: CAMs under clean and adversarial inputs.
- **Columns 4–5 (w/o ECA)**: CAMs for the same architecture without ECA.

We observe that models with ECA tend to produce more focused, class-consistent activation maps. For instance, in the frog example (row 3), clean and adversarial CAMs remain centered on the object when ECA is enabled, whereas the w/o-ECA counterpart exhibits a pronounced shift that coincides with a misclassification to "dog." In the cat example (row 4), the ECA model preserves attention over the cat body, while the w/o-ECA variant misclassifies it as "ship" and shifts attention to background. Similar patterns appear in the horse sample (row 5), suggesting improved stability of attention under perturbations.

Overall, these qualitative results indicate that ECA can promote both adversarial robustness and interpretability by enforcing more discriminative and stable attention over semantically meaningful regions.

### A.3 HYPERPARAMETER SETTINGS AND SEARCH DETAILS

To ensure fair comparison and reproducibility, we follow standard settings widely adopted in robust NAS for optimization and adversarial attacks. Unless otherwise noted, all experiments use:

- **Optimizer**: SGD with momentum 0.9, initial learning rate 0.025, cosine annealing;
- **Weight decay**: $3 \times 10^{-4}$;
- **Search epochs**: 50;
- **Adversarial training**: PGD-7 with step size $2/255$ and $\epsilon = 8/255$ (untargeted, random starts; evaluation protocol in Sec. 4.2).

We only tune the two hyperparameters introduced by the **ROSE** module:

- $\beta$: trade-off coefficient between the median-of-means (steady contribution) and the IQR-based outlier score (occasional indispensability);
- $\lambda = (\lambda_1, \lambda_2)$: asymmetric IQR scale factors for upper/lower fences, used to detect significant outliers in clean/adv gains, respectively, with

$$\tau^+ = Q_3 + \lambda_1 \, \mathrm{IQR}, \qquad \tau^- = Q_1 - \lambda_2 \, \mathrm{IQR}.$$

We conduct a lightweight grid search over:

- $\beta \in \{0, 0.2, 0.3, 0.4, 0.5, 1\}$;
- $(\lambda_1, \lambda_2) \in \{1.2, 1.5, 2.0\} \times \{1.2, 1.5, 2.0\}$.

Table 6 summarizes CIFAR-10 results under different $(\lambda_1, \lambda_2)$ choices. The best-performing setting is $(\lambda_1 = 1.2, \lambda_2 = 2.0)$, which yields strong clean accuracy and adversarial robustness.

### A.4 CELL VISUALIZATION OF THE FINAL ARCHITECTURE

To enhance reproducibility and provide architectural insights, we visualize the three cell types discovered by RDNAS on CIFAR-10: the *Normal* cell, the *Reduction* cell, and the *Robust* cell. Each cell is a directed acyclic graph in which nodes denote intermediate feature maps and edges denote selected operations. Inputs $c_{k-2}$ and $c_{k-1}$ are the outputs of the two preceding cells, and $c_k$ is the current cell's output. The final genotype is obtained from the discretized architecture at the end of search.

Table 6: CIFAR-10 performance under different $(\lambda_1, \lambda_2)$ for asymmetric IQR thresholds. Best is **bold**, second best is underlined. Attacks follow Sec. 4.2.

| $\lambda_1$ | $\lambda_2$ | Clean | FGSM | PGD$^{20}$ | PGD$^{100}$ |
|---|---|---|---|---|---|
| 1.5 | 1.2 | 84.69% | 58.41% | 50.45% | 50.07% |
| 1.5 | 1.5 | 85.45% | 59.55% | 52.18% | 51.73% |
| 1.2 | 2.0 | **86.56%** | **60.44%** | **52.62%** | **52.24%** |
| 2.0 | 1.5 | 85.05% | 57.74% | 49.97% | 49.61% |

Table 7: CIFAR-10 performance under different $\beta$ values. Best is **bold**. Attacks follow Sec. 4.2.

| $\beta$ | Clean | FGSM | PGD$^{20}$ | PGD$^{100}$ |
|---|---|---|---|---|
| 0 | 85.25% | 60.21% | 52.55% | 52.08% |
| 0.2 | 85.06% | 58.68% | 50.54% | 50.20% |
| 0.3 | **86.56%** | **60.44%** | **52.62%** | **52.24%** |
| 0.4 | 85.49% | 59.63% | 52.22% | 51.67% |
| 0.5 | 84.83% | 58.48% | 50.99% | 50.47% |
| 1 | 84.45% | 59.47% | 52.45% | 52.13% |

- **Normal cell.** Prioritizes expressive operations (e.g., `sepconv-3x3`, `sepconv-5x5`), forming deeper computation paths while maintaining gradient flow with occasional `skipconnect`.

- **Reduction cell.** Responsible for spatial downsampling, often combining pooling (e.g., `maxpool-3x3`) with separable convolutions to expand the receptive field efficiently.

- **Robust cell.** Tends to favor smoother or more redundant topologies (e.g., `dilconv-5x5`, `skipconnect`), which are associated with improved stability under perturbations.

Together, the three cells emphasize complementary goals—expressiveness (Normal), efficiency (Reduction), and stability (Robust)—which helps explain the observed balance between clean accuracy and adversarial robustness.

## A.5 Full-Scale Visualization of Primitive Trends

To provide a comprehensive view of search dynamics, we visualize the high-resolution trigger trends of each primitive in both the *Normal* and *Robust* cells across 50 search epochs. While the main paper provides compact summaries for space, the figures below reveal finer-grained patterns and fluctuations. For clarity, "exception triggers" (method-specific events such as IQR outliers) are counted per epoch; see Sec. 3.4 for the formal definition.

We observe that primitives such as `sepconv-3x3/5x5` and `dilconv-5x5` are *often* among the most frequently triggered in both cells, suggesting their importance for robust architecture formation. In contrast, `skipconnect` and pooling operations (e.g., `avgpool-3x3`, `maxpool-3x3`) are triggered less frequently, indicating a lower tendency to be prioritized in our robustness-aware search space.

The temporal curves also exhibit periodic rises and falls, implying that the search revisits and re-assesses operator importance over time rather than converging prematurely. These detailed trends support the effectiveness of our ROSE-based exception mechanism in tracking meaningful shifts in primitive utility.

## A.6 Attention Module Selection

To assess how the fusion attention choice shapes both the searched architecture and robustness, we evaluate three candidates—CBAM (reduction= 16, $k$=7) (Woo et al., 2018), ECAM ($k_c$=3, $k_s$=5) (W et al., 2025), and ECA ($k$=3) (Wang et al., 2020; Zhang et al., 2025)—by replacing the attention module in the search space and re-running the full NAS pipeline for each variant. Apart

Table 8: Attention module comparison on CIFAR-10 (%, $\ell_\infty$, $\epsilon = 8/255$, PGD step size 2/255). Single run per setting; best in each column is **bold**.

| Module | Params (M) | Clean | FGSM | PGD$^{20}$ | PGD$^{100}$ | APGD$_{\text{CE}}$ |
|---|---|---|---|---|---|---|
| CBAM (reduction=16, k=7) | 4.2 | 84.62 | 59.72 | 52.38 | 51.99 | 51.73 |
| ECAM ($k_c = 3$, $k_s = 5$) | 4.5 | 85.34 | 59.92 | 52.57 | 52.17 | 51.95 |
| **ECA** ($k = 3$) | 4.4 | **86.56** | **60.44** | **52.62** | **52.24** | **52.05** |

from the attention module, all factors are held fixed: the rest of the primitives, random seeds and splits, search/training budgets, and optimization hyperparameters (including 7-step PGD adversarial training with step size 2/255, $\epsilon$=8/255).

As reported in Table 8, under the "re-search + retrain" setting, ECA yields the best or tied-best performance across all metrics. ECAM (a spatially-extended ECA) is close under strong attacks but slightly behind overall; CBAM drops further under stronger attacks. Because the resulting models have nearly identical parameter counts, these gains are unlikely to stem from model size, pointing instead to how attention design shapes the searched topology and robustness-oriented inductive bias.

**Why does ECA lead the search?** We believe three factors contribute: (1) No channel bottleneck: ECA's local 1D channel convolution avoids MLP reduction, preserving fine-grained discriminative information that is easily lost under adversarial training; (2) Local channel neighborhoods: small-kernel cross-channel interactions appear more stable to perturbation noise, reducing variance in Shapley-style scoring and gradient updates during search; (3) Lightweight and optimization-friendly: with minimal parameters and a short optimization path, ECA couples more benignly with adversarial losses, whereas spatial branches and strong reduction (CBAM/ECAM) add compute and can aggravate optimization instability without consistent robustness gains.

## A.7    Robustness Across PGD Budgets

To examine how robustness evolves with the strength of the adversary, we sweep the perturbation budget $\epsilon$ for untargeted $L_\infty$ PGD on CIFAR-10 and evaluate all models under a fixed, transparent protocol:

- **Attack setup.** For each $\epsilon \in \{1, 2, 3, 4, 6, 8\}/255$ we use step size $\alpha = \epsilon/4$, $r = 5$ random restarts with uniform initialization in the $\ell_\infty$ ball, and $T$ PGD iterations. To avoid under-optimized attacks at small budgets, $T$ is increased as $\epsilon$ decreases (Table 9). No attack parameter is tuned per model.

- **Evaluation.** Robust accuracy is measured on the full test set; images are clipped to $[0, 1]$ after each step. The same settings are applied to all compared methods.

**Results and takeaways.** Table 9 reports the robust accuracy grid. As expected, accuracy decreases monotonically as $\epsilon$ grows. Across the full range of budgets, **RDNAS** consistently matches or exceeds baselines, with the largest margins appearing at the higher budgets (e.g., $\epsilon = 8/255$). At moderate budgets ($\epsilon \in [3, 6]/255$), RDNAS preserves a comfortable robustness headroom while maintaining competitive clean accuracy (see main text), indicating that its dual-branch design and ECA fusion do not rely on a narrow operating point.

All runs use identical preprocessing, evaluation code, and random seeds across models. We follow the common choice $\alpha = \epsilon/4$ and keep restarts fixed ($r = 5$) to balance attack strength and runtime. This grid can serve as a drop-in stress test for future methods to report budget sensitivity under a standardized PGD protocol.

Table 9: Robust accuracy (%) under $L_\infty$ PGD with varying budgets. Default $\alpha = \epsilon/4$, restarts $r = 5$.

| $\epsilon$ | $\alpha$ | $T$ | $r$ | **RDNAS(Ours)** | **LRNAS** | **RACL** | **AdvRush** |
|---|---|---|---|---|---|---|---|
| 8/255 | 2/255 | 20 | 5 | **52.62** | 50.20 | 51.89 | 52.29 |
| 6/255 | 1.5/255 | 20 | 5 | **62.84** | 56.05 | 61.94 | 62.09 |
| 4/255 | 1/255 | 20 | 5 | **72.11** | 66.45 | 71.33 | 71.94 |
| 3/255 | 0.75/255 | 30 | 5 | **76.08** | 71.05 | 75.33 | 75.49 |
| 2/255 | 0.5/255 | 40 | 5 | **79.91** | 75.12 | 79.87 | 79.70 |
| 1/255 | 0.25/255 | 50 | 5 | **83.34** | 79.29 | 82.24 | 83.14 |

## A.8 ROBUSTNESS ACROSS VARIOUS SAMPLE TYPESON CIFAR-10 UNDER $\ell_\infty$ ATTACKS

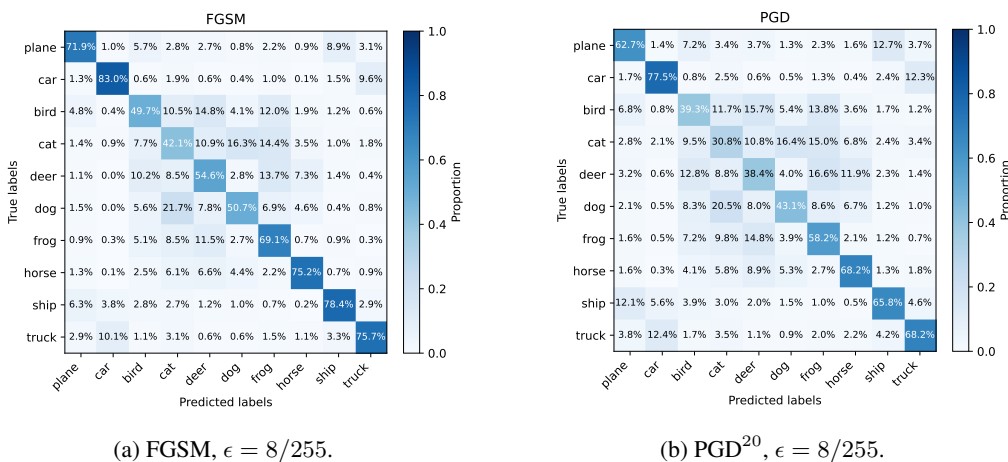

(a) FGSM, $\epsilon = 8/255$.      (b) PGD$^{20}$, $\epsilon = 8/255$.

Figure 7: Class-wise confusion matrices of the final RDNAS model on CIFAR-10 under white-box $\ell_\infty$ attacks.

To better understand why a robust estimator like ROSE is needed in our adversarial NAS setting, we report class-wise confusion matrices of the final RDNAS model on **CIFAR-10** under **white-box** $\ell_\infty$ **attacks** in Figure 7.

**Dataset:** CIFAR-10 test set (10,000 images, 10 classes).
**Model:** The final RDNAS architecture adversarially trained
**Attacks:**

- **FGSM** ($\ell_\infty$) with perturbation budget $\epsilon = 8/255$.
- **PGD$^{20}$** ($\ell_\infty$) with $\epsilon = 8/255$, step size 2/255, 20 steps, and random starts.

Figure 7 reveals a clear heterogeneity across classes: some classes (e.g., *bird, cat, deer, dog*) suffer substantial drops in diagonal accuracy and much larger mass spread over incorrect labels, whereas others (e.g., *plane, car, frog, horse, ship, truck*) remain comparatively robust under the same attack budget. This behavior indicates that, even at a fixed $\ell_\infty$ budget on CIFAR-10, the induced adversarial loss distribution is **highly skewed and class-dependent**. A relatively small subset of "hard" examples and vulnerable classes contributes disproportionately to the gradients, leading to **heavy-tailed and occasionally extreme** marginal gains when estimating operation-level Shapley values in a weight-sharing supernet. In such a regime, naive averaging of marginal gains can be dominated by these outliers and yield unstable operation rankings across runs. ROSE explicitly addresses this issue by combining (i) a **Median-of-Means** term, which provides a robust estimate of the typical contribution under heavy-tailed noise, and (ii) an **IQR**-based component, which controls the influence of rare but extreme deviations. The CIFAR-10 confusion matrices in Figure 7 thus provide empirical motivation for using ROSE as a robust operation-scoring mechanism in adversarial NAS.

## A.9 ADDITIONAL IMAGENET-1K EXPERIMENTS

Following TRNAS (Yang et al., 2025), we train all architectures on ImageNet-1k with FAST-FGSM adversarial training for 50 epochs, perturbation radius $\epsilon = 4/255$, and SGD with momentum. We use a batch size of 512 and apply the same training protocol to DARTS (Liu et al., 2019), LR-NAS (Feng et al., 2025a), CRoZe (Ha et al., 2023), ZCPRob (Feng et al., 2025b), TRNAS, and our searched architecture. After training, we report top-1 accuracy on clean images, FGSM, and PGD-20 attacks.

Table 10: Comparison of NAS methods on ImageNet-1k under FAST-FGSM training ($\epsilon = 4/255$). We report top-1 accuracy (%).

| Methods | Type | Clean | FGSM | PGD-20 |
|---------|------|-------|------|--------|
| DARTS | Clean | 53.88 | 19.22 | 11.11 |
| LRNAS | Robust | 48.21 | 15.02 | 8.18 |
| CRoZe | Zero-shot robust | 49.52 | 16.28 | 9.41 |
| ZCPRob | Zero-shot robust | 52.93 | 18.86 | 10.75 |
| TRNAS | Zero-shot robust | 55.10 | 20.56 | 11.73 |
| Ours | Robust | **57.09** | **22.11** | **12.68** |

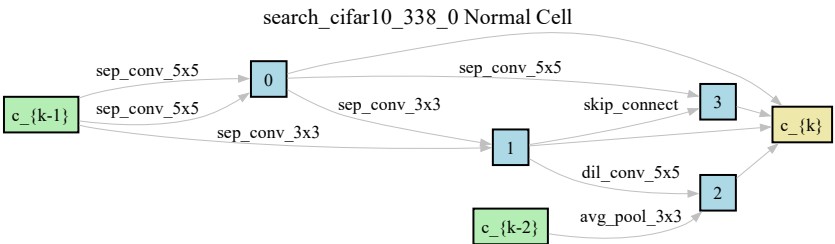

(a) Normal cell discovered by RDNAS on CIFAR-10.

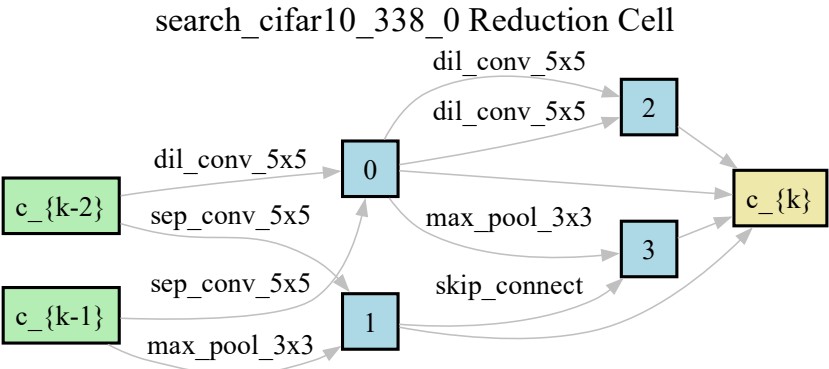

(b) Reduction cell discovered by RDNAS on CIFAR-10.

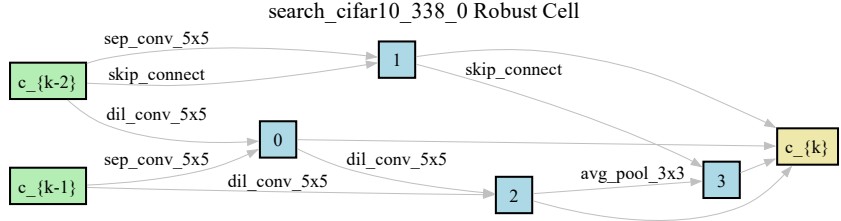

(c) Robust cell discovered by RDNAS on CIFAR-10.

Figure 4: Cells discovered by RDNAS on CIFAR-10. From top to bottom: Normal, Reduction, and Robust cells.

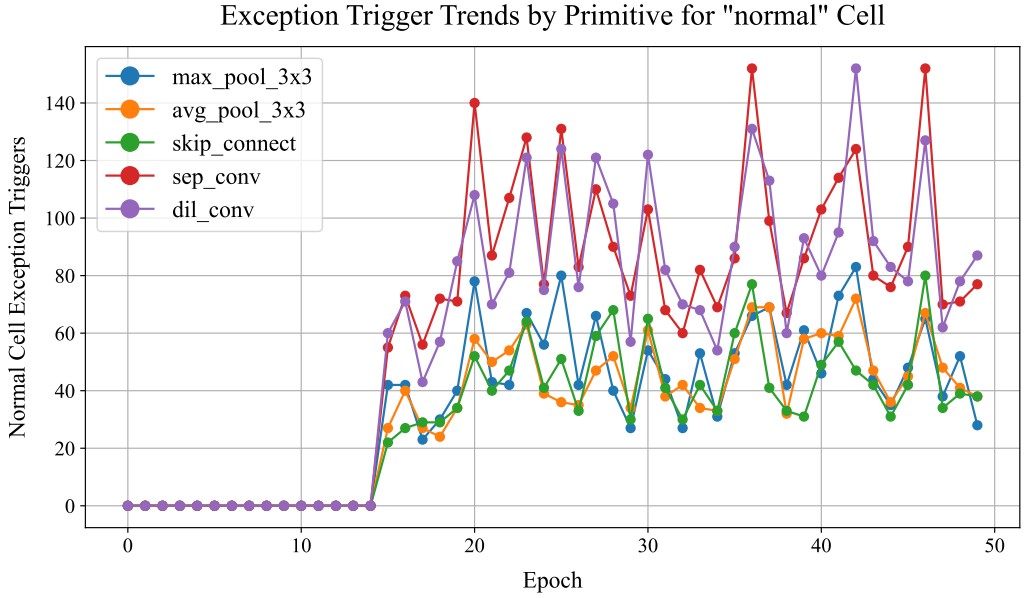

Figure 5: High-resolution trigger trends for primitives in the Normal cell across 50 search epochs.

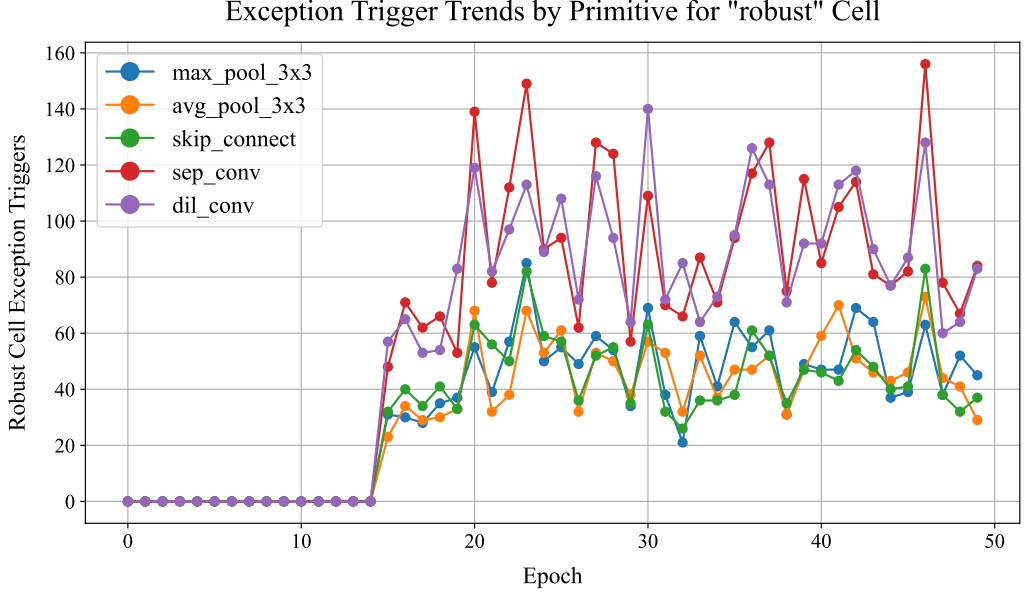

Figure 6: High-resolution trigger trends for primitives in the Robust cell across 50 search epochs.

