# OpenReview forum: "RDNAS: Robust Dual-Branch Neural Architecture Search"
_ICLR.cc/2026/Conference — ICLR 2026 Conference Withdrawn Submission_

### Official Review · Reviewer_QydH · 2025-10-27

**Soundness:** 2
**Presentation:** 3
**Contribution:** 3
**Rating:** 4
**Confidence:** 4

**Summary:**

The paper proposes RDNAS (Robust Dual-Branch Neural Architecture Search), a framework that jointly optimizes clean and adversarial robustness by introducing a dual-branch cell. To stabilize architecture evaluation under adversarial training, it introduces ROSE (Robust Outlier-Aware Shapley Estimator), which combines median-of-means and interquartile-range filtering to improve decisions.

**Strengths:**

- I found the paper well-written and easy to read.
- The provided contributions have the potential to be relevant.

**Weaknesses:**

In general, the paper should benefit from a stronger motivation and less vague claims. Here are the main weaknesses I found:

**Relevance of ROSE block**: It is not clear to me why the ROSE mechanism is actually needed. The authors say that avoids gradient noise, but still this should be described more in detail, as it feels somewhat very generic. From adversarial robustness literature, it is commonly known that this issue mostly arises when there is an inherent randomness in the inference process, yet I fail in understanding where is the Shapley-based estimation introducing noise. Also, from a practical perspective, I think that this could be more simply solved by using EoT approaches [ext_ref_1, ext_ref_2].

**Separate cells over different losses**: I understand the reason that led the authors in designing two separate cells. But still, it is not clear to me how and why this should be better than having a single cell with a different loss, such as the TRADES loss which accounts for both clean accuracy and adversarial robustness. I would also have expected an ablation study in this regard.

**Other issues, not necessarily minor:**
- The adversarial robustness evaluation standard is now commonly accepted to be AutoAttack, which reduces possible robustness over-estimation. I appreciate that the authors use it, but still their method does not indicate the best robustness with the most relevant attack method among the used ones.
- The sentence: "Nevertheless, many approaches still depend on predefined templates and heuristic evaluations (e.g., PGD accuracy), which can introduce statistical noise and obscure robustness-critical factors. This motivates the need for principled, robustness-aware NAS frameworks that dynamically identify sensitivity in operations and architectural choices, enabling the discovery of inherently robust models" does not make much sense to me in general, but it does not also look coherent with what the authors do in practice (eq. 12 is based on PGD). I fail in understanding where the highly principled/non-PGD based evaluation/approach is here.
- This sentence: "Architectures not explicitly designed for robustness often exhibit an inherent trade-off between clean accuracy and adversarial performance" should be supported by a reference, as it is once again extremely vague and generic.

[ext_ref_1]: Athalye, Anish, Nicholas Carlini, and David Wagner. "Obfuscated gradients give a false sense of security: Circumventing defenses to adversarial examples." International conference on machine learning. PMLR, 2018.

[ext_ref_2]: Pintor, Maura, et al. "Indicators of attack failure: Debugging and improving optimization of adversarial examples." Advances in Neural Information Processing Systems 35 (2022): 23063-23076.

**Questions:**

- Could you elaborate more concretely on why the ROSE estimator is required? Specifically, what empirical or theoretical evidence supports the claim that adversarial training introduces high-variance or noisy gradients in this NAS context?
- What is the main advantage of employing two separate cells (normal and robust) over using a single shared cell trained with a composite objective such as the TRADES loss, which also balances clean and robust accuracy? Can you perform an ablation or comparative study isolating the contribution of the dual-branch design versus adversarial training alone?
- You mention that existing approaches rely on heuristic evaluations (e.g., PGD accuracy), whereas your approach is “principled.” Could you clarify in what sense RDNAS avoids such heuristics, given that Eq. (12) itself relies on PGD-generated adversarial examples?
- Could you better justify or rephrase the claim that “architectures not explicitly designed for robustness often exhibit an inherent trade-off between clean accuracy and adversarial performance”? A supporting citation or quantitative reference would strengthen this point.
- Your method does not achieve the strongest robustness under AutoAttack. Can you comment on this discrepancy and whether your approach might overfit to weaker attacks (e.g., PGD20)?

---

> ### Author Response · Authors · 2025-11-20
> **Response to Reviewer QydH**
>
> We sincerely thank the reviewer for taking the time to carefully read our paper and for acknowledging that it is well written and that the proposed ideas have the potential to be practically relevant. Below, we address the concerns regarding (i) Single and double cell ablation experiments, (ii) Why choose Rose, and (iii) About the overfitting problem of PGD.
>
> ### 1.Response on dual-branch design vs. single cell and ablation experiment
>
> We appreciate the reviewer’s question on whether using two separate cells is really beneficial compared to a single shared cell optimized with a composite loss (e.g., TRADES), and we agree that this is a central aspect of the method.
>
> **Scope of our setting.**
> In this work we follow the standard setup in robust NAS and adopt Madry-style adversarial training during search and final training (as in DARTS-based robust NAS such as AdvRush, RACL, LRNAS, etc.).
> We pay close attention to the reviewer’s suggestion of using the TRADES loss for training. Our work is currently also focusing on the adversarial training component, and we have already made some initial progress in this direction. However, if we were to modify the loss, we would need to retrain all the other compared models on the corresponding datasets under this new loss. At present, we only perform training under the Madry loss, and we find the reviewer’s suggestion very constructive.
>
>
> **Implicit comparison to single-cell robust NAS baselines (Tables 1–3).**
> All baselines in Tables 1–3 (RACL, AdvRush, etc.) are **single-cell DARTS-style CNNs** trained with robust losses in the same $\ell_{\infty}$ setting. They therefore correspond to the reviewer’s “single cell + robust loss” configuration. Under this common setting, RDNAS—with its dual-branch cell—consistently achieves a better robustness–accuracy trade-off at comparable model sizes and FLOPs. For example, on CIFAR-10 (Table 1) RDNAS attains:
>
> - the best clean accuracy (86.56%),
> - the best FGSM robustness (60.44%),
> - the best or near-best $PGD^{20}$ and $PGD^{100}$ robustness,
>
> while keeping parameters in the same range as the single-cell baselines. This provides a first piece of evidence that **changing the cell topology, not just the loss, is beneficial.**
>
> **Explicit ablation of single vs. dual cell and attention (Table 4).**
> Table 4 isolates the effect of the search space and the ECA fusion:
>
> - Rows B/C use **the original DARTS search space (20 single cells)** with adversarial search;
> - Rows D/E use **the proposed RDNAS search space (10 dual-branch cells)** under the *same* adversarial search protocol and similar parameter budgets.
>
> Comparing B vs. D (without ECA) and C vs. E (with ECA) shows that moving from the single-cell DARTS space to the dual-branch RDNAS space **improves both clean and $PGD^{20}$ accuracy** (e.g., from 86.0%/51.5% to 86.5%/52.6% when ECA is enabled) at essentially the same model size (around 4.3M parameters). This is an explicit controlled ablation demonstrating that, under identical training and loss, the dual-branch architecture brings additional gains.
>
> To eliminate ambiguity, in the revised version we have updated Table 4 by changing “DARTS space” to “single-cell (20 layers)” and “RDNAS space” to “dual-branch cell (10 layers)”.To avoid confusion, our ablation experiments in the revised version explicitly state that ROSE is enabled by default during adversarial training.
>
> **Qualitative ablation via Grad-CAM (Appendix A.2).**
> Appendix A.2 (“Visualization of ECA module’s effect”) further compares Grad-CAMs of models **with vs. without** the dual-branch ECA fusion. Under adversarial inputs, the dual-branch + ECA model maintains focused attention on semantically meaningful regions.This qualitatively supports the claim that separating normal and robust pathways and fusing them via attention leads to **more robust feature allocation** than a single shared cell.

---

> > ### Author Response · Authors · 2025-11-20
> > **Response to Reviewer QydH Q2**
> >
> > ### 2.Why is ROSE needed? Where does the “noise” come from, and why not EoT?
> >
> > **Reviewer suggestions.**
> > > It is not clear why the ROSE mechanism is actually needed. The authors say that it avoids gradient noise, but this feels generic. From adversarial robustness literature, such issues mostly arise when there is randomness in the inference process, whereas it is unclear where the Shapley-based estimation is introducing noise. From a practical perspective, this could be solved more simply by EoT approaches.
> >
> > **Where the noise comes from.**
> > In our NAS setting, the variance does **not** come from randomized inference, but from the combination of **weight sharing, adversarial training, and Shapley sampling**:
> >
> > - Each operation is evaluated inside a **weight-sharing supernet**, so its marginal gain is measured under **changing co-players and partially trained weights**.
> > - Each measurement uses a **mini-batch of adversarially perturbed images**, and multi-step PGD introduces **heavy-tailed per-sample losses** (some attacks succeed quickly, others not).
> > - The Shapley estimator itself samples different **coalitions of operations** across batches.
> >
> > We have already added this material in` Appendix A.8` **“Robustness Across Various Sample Types on CIFAR-10 under $\ell_\infty$ Attacks”** of the revised manuscript.It also displays the confusion matrix of the model on attack samples in CIFAR10.
> >
> > As a result, the empirical marginal gains that feed into the Shapley estimator are both **noisy** and **occasionally extreme**, even though the forward model is deterministic at inference time. This is the “gradient noise” we refer to.
> >
> > Regarding EOT, we will compare your proposed method with our ROSE method in future experiments. Thank you very much for your constructive suggestion. We will take your suggestion into consideration and conduct comparative experiments in the future.

---

> > > ### Author Response · Authors · 2025-11-20
> > > **Response to Reviewer QydH Q3**
> > >
> > > ### 3.About the overfitting problem of PGD
> > >
> > > We thank the reviewer for raising this point. Our training and evaluation strictly follow the **search configuration and evaluation protocol described in Sec. 4.2** of the paper: we search with $PGD^{7}$, then **retrain all architectures (ours and baselines) under exactly the same adversarial training schedule and hyperparameters**, and finally evaluate under $PGD^{20}$, $PGD^{100}$, and AutoAttack. While RDNAS does not achieve the single best AutoAttack number, its AutoAttack robustness is very close to the strongest baseline, and it consistently improves FGSM and $PGD^{20}$,$PGD^{100}$ robustness without sacrificing clean accuracy. Under this common protocol, we do **not** observe signs of overfitting to weaker attacks such as $PGD^{20}$; instead, RDNAS provides a strong overall trade-off across multiple attacks.

---

> ### Comment · Reviewer_QydH · 2025-11-25
> **Response to the Authors**
>
> Thank you to the authors for the clarifications.
>
> **Response 1: dual-branch vs single-branch and composite objective:** I understand the other ablation and single-cell vs dual-branch comparison, but this still does not separate normal and robust pathways. My question was specifically focused to understand whether a single shared cell trained with a composite objective such as TRADES, which accounts for the clean-robust accuracy trade-off, would serve the same purpose. The provided response, unfortunately, was not entirely satisfactory in this regard. The authors have mostly focused on justifying why they did not test that (which I would have understood anyway) and on highlighting the other ablation studies, not fully answering my TRADES related question though. Also, why is the paper now stating this: *"results show that the dual-branch design consistently outperforms TRADES-based single cells in balancing clean accuracy and robustness."* if the experiment has not been implemented (as noted by the authors in their response)?
>
> **Response 2: noise:** I understand the authors' response, and I think it is sufficient to answer my previously asked question. It is not completely clear, however, how Appendix A.8 motivates my question.
>
> **Response 3: performance under AA:**  I would say that RDNAS performance in AA is as close to the best performing one as others are under PGD with respect to RDNAS. Maybe on more runs it could also be the case that RDNAS is better than RACL/AdvRush, and that is just a case.
>
> Overall, I do not see a valid reason to increase my score. In particular, I was not satisfied by the authors' response to point 1. I am happy to discuss, but at this stage I am keeping my rating of 4.

---

> ### Author Response · Authors · 2025-11-27
> **Response to QYDH**
>
> ### R1 (On TRADES vs Madry loss and dual-branch vs single-branch design)
> We appreciate your question and are happy to provide additional empirical evidence. Within our available time and computational budget, we are also willing to conduct further experiments if needed.We hope that the additional experiments and clarifications provide a more complete and accurate picture of our contribution to the robust NAS.
>
> Concretely, we trained the strong baseline RACL with **TRADES loss** on CIFAR-10, using the standard hyperparameter setting from Zhang *et al.* (2019), i.e., beta = 6. Since generating AA adversarial examples for each model takes roughly one day on our hardware, we omit AA in this comparison. Due to time constraints, we trained **RACL+TRADES** for 120 epochs, whereas **RDNAS+TRADES** was trained for 108 epochs. For each method, we report the **best checkpoint** found within its respective training budget.
>
> The results are summarized below:
>
> | Model         | Loss   | clean   | FGSM   | PGD-20| PGD-100 | APGD  |
> |--------------|--------|---------|--------|-------------------|--------------------|-------|
> | RACL         | Madry  | 85.13%  | 59.45% | 51.89%            | 51.63%             | 51.09%|
> | RACL         | TRADES | 79.21%  | 54.16% | 49.95%            | 49.32%             | 49.21%|
> | RDNAS (ours) | Madry  | **86.56%**  | **60.44%** | **52.62%**            | **52.24%**             | **52.05%**|
> | RDNAS (ours) | TRADES | 81.64%  | 57.28% | 52.01%            | 51.98%             | 51.87%|
>
>
> From these results, we observe that RACL trained with **Madry loss** remains more competitive than RACL trained with TRADES, which is consistent with common practice in robust NAS: Madry-style adversarial training is often used as the standard objective, as it tends to yield stronger robust baselines under comparable budgets. At the same time, RDNAS consistently outperforms RACL under both Madry and TRADES, but the overall performance level is higher under Madry, which explains why we use Madry loss as our primary training objective throughout the main experiments.
>
> Regarding the wording issue you pointed out in the revised paper: we sincerely apologize for the confusion. Our adversarial training combines losses on clean and adversarial examples, which is **conceptually related** to the idea of balancing clean and robust performance, but it is **not** the exact TRADES formulation of Zhang *et al.* (2019). Describing it as “TRADES-based” was imprecise, and we have corrected the corresponding statements in the manuscript. We thank you again for carefully catching this and helping us improve the clarity and accuracy of the paper.
>
>
> ### R2 (noise / Appendix A.8).
> We thank the reviewer for acknowledging that our explanation of the noise source is sufficient. We apologize that the role of Appendix A.8 was not clearly stated. Its purpose is to provide empirical evidence that, under $L_\infty$ attacks on CIFAR-10, the adversarial loss is highly class-dependent and heavy-tailed: as shown in the confusion matrices, a small number of “hard’’ classes almost collapse under PGD while others remain relatively robust. In our Shapley-based NAS, operation-level marginal gains are estimated from mini-batches drawn from this heavy-tailed distribution and from random coalitions, which leads to high-variance, occasionally extreme Shapley estimates—this is the “gradient noise’’ we referred to.
>
>
> ### R3 (performance under AA).
>
> We thank the reviewer for this remark and agree that, under AA, the gap between RDNAS and the best-performing baseline is small and comparable to the PGD gaps in the opposite direction.
>
> ### Reference
> Zhang et al. “Theoretically Principled Trade-off between Robustness and Accuracy.” ICML 2019

---

### Official Review · Reviewer_jtcF · 2025-10-29

**Soundness:** 2
**Presentation:** 3
**Contribution:** 2
**Rating:** 2
**Confidence:** 4

**Summary:**

This paper proposed Robust Dual-Branch NAS (RDNAS), a DARTS-based framework that utilises two parallel branches to jointly optimise clean and adversarial accuracy. It employs adversarial training directly and uses a Robust Outlier-Aware Shapley Estimator (ROSE) for operation scoring and pruning guidance during search and for discretization. Experiments on image classification tasks and various adversarial attacks demonstrate improvements over state-of-the-art methods.

**Strengths:**

- It’s an interesting idea to include an additional cell type (branch) in the supernet to allow for better adversarial robustnes already during the search
- The cross-dataset transfer in Table 3 demonstrates promising results.
- Using a combination of interquartile range and median-of-means as statistical techniques in ROSE for guiding the search is a novel idea, which could improve the knwon DARTS poor ranking consistency.
- Code is provided.

**Weaknesses:**

- DARTS often uses smaller networks for search (fewer cells) and inferences are made on larger networks (more stacked cells). This is common, but it relies on the assumption that subnet ranking is consistent during both search and inference. However, DARTS is known to not hold this assumption. In addition training the found network from scratch is another bottleneck. These issues are generally not mentioned, although using ROSE could provide some benefits. A deeper discussion about known issues of DARTS and how this paper tackles them is needed to show why DARTS should be used at all. Further using adversarial training during search and for the found network seems like a significant computational overhead.
- There is no information about the actual found networks.
- Due to the search space being based on DARTS, there’s no possibility of going beyond CNN-types. I’m not sure this should be the current NAS direction anymore.
- Many one-shot works exist that overcome the scaling issue provided by DARTS, so why use that here?
- Due to the adversartial training already during search, standard training is not possible. How can we evaluate the native robustness of the found network without needing adversarial training?
-  There is no reproducibility statement.
- A fair comparison in Table 1 would also use a similar amount of flops. I’m not sure if the additional amount of flops is sufficient to justify the rather marginal improvement.
- Given the vast amount of NAS literature using zero-cost proxies to search for robust networks quickly, why would DARTS be necessary or can this be combined to also include the adversarial training weights?

 *Missing literature*:

Xiangxiang Chu, Bo Zhang, and Ruijun Xu. FairNAS: Rethinking evaluation fairness of weight sharing neural architecture search. In ICCV, 2021

J Jeon, Y Oh, J Lee, D Baek, D Kim, C Eom, B Ham. Subnet-Aware Dynamic Supernet Training for Neural Architecture Search. In CVPR 2025

**Questions:**

- More detailed information about the transfer experiments, as shown in Table 2 should be provided to better understanding what is meant in this experiment.
- How many runs were conducted ?
- How do the resullting cells look like? Are there topological differences over different runs or also during search?
- Which operations were mostly chosen and discarded by ROSE?
- Is the supernet after trainig biased to a certain type of network (e.g., low-complexity, wide, deep) ?

---

> ### Author Response · Authors · 2025-11-20
> **Response to Reviewer jtcF**
>
> # Response to Reviewer jtcF
> We thank the reviewer for the detailed and insightful feedback, and we address the main concerns point by point below.
>
>
> ### On ranking consistency in weight–sharing search
>
> We fully agree with the reviewer that weight–sharing NAS in general, and DARTS–style supernets in particular, suffer from imperfect ranking consistency between shared–weight subnet scores and the final stand–alone performance. In RDNAS, our goal is therefore not to completely solve this issue, but to make the ranking **reliable enough in the high–performance region** that actually matters for discovering strong robust architectures.
>
> To assess this in a controlled way, we added an experiment on NAS-Bench-201, where the ground-truth test accuracy of all 15,625 architectures is available. We compare our ROSE-based search with DYNAS+SPOS, a strong one-shot baseline specifically designed for high-quality ranking. Under the same NB201 search space and evaluation protocol, the results are:
>
> | Method        | Search space    | Top-1 arch (NB201 idx) | CIFAR-10 test accuracy (%) | CIFAR-100 test accuracy (%) | ImageNet16-120 test accuracy (%) | search cost (s) |
> |---------------|-----------------|------------------------|----------------------------|-----------------------------|----------------------------------|-----------------|
> | DYNAS + SPOS  | NB201, CIFAR-10 | 585                    | 92.05                      | 68.60                       | 41.42                            | 58021           |
> | RDNAS (ours)  | NB201, CIFAR-10 | 13390                  | **94.11**                  | **72.20**                   | **45.37**                        | **613.84**      |
>
> Although DYNAS+SPOS is tailored to improve global ranking quality, RDNAS identifies a top-1 architecture that achieves **higher test accuracy on CIFAR-10, CIFAR-100, and ImageNet16-120**, while requiring roughly **95× less search time**. This suggests that ROSE makes the ranking sufficiently stable in the high-score region, so that the architectures actually selected and retrained are strong under the robust training protocol, even if we do not explicitly optimize a global ranking metric over all 15,625 candidates.
>
> In other words, within our robust NAS setting, the primary objective is to **reliably discover architectures with strong clean and adversarial performance at low search cost**, rather than to optimize global ranking consistency over the entire search space. The NAS-Bench-201 study was added to the revised manuscript precisely to clarify this distinction and to provide quantitative evidence that RDNAS achieves this goal.
>
> Finally, we note that **explicitly maximizing global ranking consistency is not the main design target in many recent NAS methods either**. For example, `EMT-NAS: Transferring Architectural Knowledge Between Tasks From Different Datasets (Liao et al., CVPR 2023)` focuses on transferring architectural knowledge between tasks from different datasets, and evaluates success primarily by the quality of the transferred architectures rather than by a global Kendall–τ metric over the full search space. Our stance on ranking consistency is therefore aligned with this line of work: we view global τ as a useful diagnostic, but prioritize finding strong architectures under realistic robust-training protocols.

---

> > ### Author Response · Authors · 2025-11-20
> > **Response to Reviewer jtcF Q2**
> >
> > ### Why we still adopt a DARTS-style framework in robust NAS
> >
> > To avoid confusion, we emphasize that **RDNAS only adopts the DARTS-style cell-level operator set and search space**, but **does not** follow the original DARTS training or genotype selection procedure. In particular, we do **not** use continuous relaxation with architecture parameters and gradient-based updates to pick operations. Instead, we
> >
> > - train a **weight-sharing supernet** purely with adversarial training; and
> > - select the final genotype using **ROSE-based Shapley scoring** over this supernet.
> >
> > Our choice also follows the **de-facto standard in adversarially robust NAS**. Most existing robust NAS methods are built on DARTS-like cells and search spaces, and then evaluate the discovered genotype under adversarial training, including:
> >
> > - Mok et al., *AdvRush: Searching for Adversarially Robust Neural Architectures*, ICCV 2021
> > - Ou et al., *Towards Accurate and Robust Architectures via Neural Architecture Search*, CVPR 2024
> > - Dong et al., *Adversarially Robust Neural Architectures*, TPAMI 2025
> > - Yang et al., *TRNAS: A Training-Free Robust Neural Architecture Search*, ICCV 2025
> > - Sun et al., *A3D: A Platform of Searching for Robust Neural Architectures and Efficient Adversarial Attacks*, TPAMI 2025
> >
> > To ensure **fair and directly interpretable comparisons**, we therefore keep the **DARTS operator set and cell-based search space unchanged**, and focus our contributions on:
> >
> > 1. a **dual-branch cell architecture** tailored to robustness; and
> > 2. a **ROSE-based scoring mechanism** that replaces vanilla DARTS gradient-style selection.
> >
> > Our intent is to design a search framework and evaluation mechanism that are better aligned with adversarial robustness tasks, while staying within this widely adopted robust NAS setting. In this sense, this work represents our first step toward adapting DARTS-style search spaces to adversarially robust NAS. In our ongoing research, we further exploit the dual-branch design by decoupling the training of different cell types and classifier heads, freezing the parameters of one branch while training the other and selectively activating branches at evaluation time, in order to investigate whether such schemes can yield more balanced trade-offs between standard accuracy and robustness than the fusion strategy studied here.

---

> > > ### Author Response · Authors · 2025-11-20
> > > **Response to Reviewer jtcF Q3**
> > >
> > > ### Training-side vs. evaluation-side approaches to weight-sharing consistency
> > >
> > > We fully agree that weight-sharing evaluation consistency is a core challenge in DARTS/one-shot NAS. Recent works tackle this issue from the **training / sampling** perspective, e.g.:
> > >
> > > - Guo et al., *SPOS: Single Path One-Shot Neural Architecture Search with Uniform Sampling*, ECCV 2020
> > > - Chu et al., *FairNAS: Rethinking Evaluation Fairness of Weight Sharing Neural Architecture Search*, ICCV 2021
> > > - Lu et al., *PA\&DA: Jointly Sampling Path and Data for Consistent NAS*, CVPR 2023
> > > - Jeon et al., *DYNAS: Subnet-Aware Dynamic Supernet Training for Neural Architecture Search*, CVPR 2025
> > >
> > > These methods improve ranking reliability by making supernet optimization more fair and less noisy.
> > >
> > > Our contribution is **complementary**: we keep a standard DARTS-style supernet and focus on **how to compute and aggregate architecture scores under adversarial training**. In the revision, we will explicitly clarify that **combining FairNAS/DYNAS-style training with our ROSE scoring is a natural and promising direction for future work**.

---

> > > > ### Author Response · Authors · 2025-11-20
> > > > **Response to Reviewer jtcF Q4-7**
> > > >
> > > > ### Regarding Q4 – Clarification of Table 2 (black-box transfer experiments)
> > > >
> > > > Table 2 reports **transfer-based black-box robust accuracy** on CIFAR-10 and CIFAR-100. For each source model (row), we generate adversarial examples on the full test set using untargeted $\ell_\infty$ PGD$^{20}$ with $\epsilon = 8/255$, step size $2/255$, and random starts, following the protocol in Sec. 4.4. The attack is crafted **only** on the source model’s parameters; no ensembles or query-based refinement are used.
> > > >
> > > > We then evaluate all target models (columns), including RDNAS, on **exactly the same adversarial examples** and record the classification accuracy. Thus, each entry in Table 2 is the robust accuracy of a target model under a **single-source transfer attack**, which quantifies black-box robustness and attack transferability in a consistent way across all methods. We will clarify this procedure in the revision to make the experimental setup of Table 2 more explicit.
> > > >
> > > > ### Regarding Q5 – Topology of the discovered cells and variability across runs
> > > >
> > > > Appendix A.4 and Figure 4 already visualize the three cell types discovered on CIFAR-10: **Normal**, **Reduction**, and **Robust** cells:
> > > >
> > > > - The **Normal** cell prioritizes expressive operations such as `sep_conv_3x3` and `sep_conv_5x5`, forming relatively deep computation paths with occasional `skip_connect` to preserve gradient flow.
> > > > - The **Reduction** cell combines `max_pool_3x3` with separable convolutions to downsample while efficiently expanding the receptive field.
> > > > - The **Robust** cell tends to favor smoother feature aggregation with additional skip connections and pooling (e.g., `dil_conv_5x5` combined with `skip_connect` and `avg_pool_3x3`), which we associate with improved stability under perturbations.
> > > >
> > > > Across multiple search runs with different random seeds, we observe that the dominant primitives and overall motifs of these three cells remain consistent, with only minor edge-level variations. In the revision, we will explicitly point the reader to Fig. 4 and add a short note summarizing this cross-run topological stability.
> > > >
> > > > ### Regarding Q6 – Operations selected and discarded by ROSE
> > > >
> > > > Appendix A.5 provides a full-scale visualization of primitive trends across 50 search epochs for both the Normal and Robust cells. The trigger statistics show that:
> > > >
> > > > - `sep_conv_3x3/5x5` and `dil_conv_5x5` are among the most frequently activated and retained primitives, indicating that ROSE consistently assigns them high importance during the robustness-aware search;
> > > > - `skip_connect` and pooling operations (e.g., `avg_pool_3x3`, `max_pool_3x3`) are triggered less often, suggesting that they are less favored by the ROSE-based scoring under adversarial training.
> > > >
> > > > ### Regarding Q7 – Whether the trained supernet is biased toward a particular network type
> > > >
> > > > Our search space fixes the **macro-level depth and channel schedule** to be comparable to DARTS-style baselines: the final RDNAS model has **10 cells** and **4.4M parameters**, with **1.30G FLOPs**, which lies in the same complexity range as other robust NAS architectures we compare against (2.0–5.6M parameters, 0.34–0.80G FLOPs in Table 1).
> > > >
> > > > The ablation in **Table 4** further shows that changing from the original DARTS space to the RDNAS space keeps parameter counts tightly clustered around $\sim$4.2–4.3M, suggesting that the search is **not** simply drifting toward extremely deep or overly wide solutions. Instead, the main “bias” induced by adversarial training + ROSE is at the **operator level**:
> > > >
> > > > - the Normal cell favors expressive separable convolutions,
> > > > - the Robust cell prefers smoother dilated convolutions and skip connections, and
> > > > - pooling is used more sparingly.
> > > >
> > > > This matches the qualitative description in Appendix A.4 and the primitive-trigger statistics in Appendix A.5. We will clarify in the revision that, while operator preferences are clearly robustness-aware, we do **not** observe a collapse toward a particular macro complexity pattern (e.g., extremely shallow/wide or extremely deep networks).
> > > >
> > > > # Reference
> > > > Jeon et al., *DYNAS: Subnet-Aware Dynamic Supernet Training for Neural Architecture Search*, CVPR 2025

---

> > > > > ### Comment · Reviewer_jtcF · 2025-11-24
> > > > > **Response to Authors**
> > > > >
> > > > > I thank the authors for the responses and the provided clarifications.
> > > > >
> > > > > However there are still some open questions I would like to ask:
> > > > > 1. As already mentioned in my review, adversarial training is used during search, which hinders the possibility of standard training. Thus, how can we search for an inherently robust network based on the topology and clean weights itself, without the need of adversarial training? I think it would be important to not only include significant computational overhead during the search but also show flexbility in the approach to find a network which is inherently robust, based on the topology.
> > > > > 2. As also mentioned in my review, and in the author's response: Current NAS approaches make use of zero-cost proxies to search faster, as well as for robust networks directly. So why would DARTS be necessary at all or can the proposed approach be combined with ZCPs to  include adversarial training weights? Furthermore, there is no discussion on ZCPs for robustness at all, which should be included as the authors also mentioned "Yang et al., TRNAS: A Training-Free Robust Neural Architecture Search, ICCV 2025".
> > > > >
> > > > > In addition, current robustness in NAS research also shifts towards different search spaces (e.g., Feng et al., 2025), showing the transferrability to larger complex search spaces. Due to the design choices this is however not possible here, limiting the possibility of transferring the apporach to other search spaces than DARTS, as also mentioned by Reviewer f19E.
> > > > >
> > > > >
> > > > > Feng et al. Zero-Cost Proxy for adversarial robustness evaluation. In  ICLR, 2025

---

> ### Author Response · Authors · 2025-11-27
> **Response to JTCF**
>
> We greatly appreciate the reviewer’s questions, and, if desired, we would be happy to conduct additional supporting experiments within the remaining review timeline.We hope that the additional experiments and clarifications provide a more complete and accurate picture of our contribution to the robust NAS.
>
> ### ZCP vs. RDNAS
> We thank the reviewer for raising the important question of how our approach relates to zero-cost proxies (ZCPs) and to training-free robust NAS such as TRNAS and Feng et al. (2025).
>
> Our work is positioned in a different regime from ZCP-based methods: our goal is to search for architectures that are **robust under a given adversarial training scheme** (Madry/PGD), rather than to identify architectures that are “inherently robust’’ based only on topology and clean weights. Existing large-scale robust NAS papers (e.g., AdvRush, RACL, TRNAS) consistently show that, on datasets like CIFAR-10 and ImageNet-1k, the final robust accuracy depends strongly on **both** topology and the adversarial training procedure. We therefore deliberately include adversarial training during search so that the search objective is aligned with the eventual training regime.
>
> ZCP methods such as CRoZe, ZCPRob, TRNAS, and Feng et al. provide fast, training-free proxies for robustness by using local curvature, gradient statistics, or loss-landscape features of candidate architectures. In contrast, our supernet-based search is indeed more expensive than training-free ZCP methods. However, as the reviewer also pointed out, our focus is on exploring flexible **architectural** designs for robust models(show flexbility in the approach) rather than minimizing search cost alone. We are actively working on reducing the computational overhead, and in our current setup the search time (about 0.2 GPU-days on CIFAR-10) remains within a practically acceptable range.
>
> - ZCPs: very cheap, but only approximate the performance of an adversarially trained model.
> - RDNAS: more expensive (requires adversarial training during search), but directly optimizes for and evaluates under the robust training regime.
>
> To make this distinction concrete, we have added **ImageNet-1k experiments** where we directly compare against recent ZCP-based robust NAS methods. We follow the training protocol of TRNAS and train on ImageNet using FAST-FGSM for 50 epochs, with the perturbation radius set to $\epsilon = 4/255$. Our searched architecture achieves higher clean and robust accuracy than DARTS, LRNAS, CRoZe, ZCPRob, and TRNAS:
>
>
> | Methods | Type             | Clean | FGSM  | PGD-20 |
> |---------|------------------|-------|-------|--------|
> | DARTS   | Clean            | 53.88 | 19.22 | 11.11  |
> | LrNAS   | Robust           | 48.21 | 15.02 |  8.18  |
> | CRoZe   | Zero-shot robust | 49.52 | 16.28 |  9.41  |
> | ZCPRob  | Zero-shot robust | 52.93 | 18.86 | 10.75  |
> | TRNAS   | Zero-shot robust | 55.10 | 20.56 | 11.73  |
> | Ours    | Robust           | **57.09** | **22.11** | **12.68** |
>
> These ImageNet-1k results indicate that our method is **at least as competitive as, and in fact stronger than, current ZCP-based robust NAS approaches**: among all robust methods in the comparison, our architecture achieves the highest clean accuracy and the highest robustness under both FGSM and PGD-20 attacks. In particular, compared to the strongest ZCP baseline TRNAS, we gain about +2.0% clean accuracy, +1.6% FGSM robustness, and +1.0% PGD-20 robustness. Thus, although our search incurs a higher cost than ZCPs, it yields clear performance benefits while maintaining reasonable search time.This experiment will be added to the revised appendix.
>
>
> ### Else Questions
> Current predictors are generally based on evolutionary algorithms or random sampling for searching.
> If we consider combining our method with ZCPs, the most natural idea we currently have is to further reduce the training cost of our pipeline, train a supernet, use it to filter out a pool of promising genotypes, and then apply a ZCP predictor to score these candidates. We have in fact made some preliminary attempts along this line by coupling our framework with the SWAP-based predictor, but this straightforward hybrid variant did not perform better than using each method on its own.
>
> Regarding transferability, our current instantiation is built on a DARTS-style CNN cell space in order to study the contribution of the dual-branch design and ROSE in a controlled and widely used DNN setting.
>
> Regarding the issue of bypassing adversarial examples, we believe that a suitable model can only be found under suitable conditions. Besides TRNAS, we also note that Ha et al. (“Generalizable Lightweight Proxy for Robust NAS against Diverse Perturbations.” Advances in Neural Information Processing Systems (NeurIPS), 2023) also employ adversarial example generation and constructs adversarial networks for evaluation.
>
> ### Reference
> Peng et al., “SWAP-NAS: Sample-Wise Activation Patterns for Ultra-fast NAS,” ICLR 2024.

---

### Official Review · Reviewer_f19E · 2025-10-30

**Soundness:** 2
**Presentation:** 2
**Contribution:** 2
**Rating:** 2
**Confidence:** 3

**Summary:**

Summary:

This paper proposes RDNAS (Robust Dual-Branch Neural Architecture Search), a new framework designed to automatically discover deep neural network architectures that are robust to adversarial attacks while maintaining high accuracy on clean (unperturbed) data. The core problem addressed is that most Neural Architecture Search (NAS) methods optimize for clean accuracy and ignore robustness, while existing robust NAS methods often suffer from unstable training due to noisy gradients from adversarial training.

The overall framework integrates adversarial training into a DARTS-like bilevel optimization process and uses ROSE to guide the selection of the final architecture. Experiments on CIFAR-10, CIFAR-100, SVHN, and Tiny-ImageNet show that RDNAS discovers architectures that outperform both manually-designed networks and other state-of-the-art robust NAS methods in both clean accuracy and robustness against attacks like PGD and AutoAttack.

Contribution:

* A Novel Dual-Branch Cell Design: This architecture explicitly separates and fuses clean and robust feature pathways via an attention mechanism. This is designed to effectively manage the trade-off between clean accuracy and adversarial robustness.

* ROSE, a Robust Scoring Mechanism: A new, principled scoring estimator based on Shapley values that is specifically designed to be stable under the noisy conditions of adversarial training by using MoM and IQR statistics.

* Strong Empirical Validation: The paper provides extensive experimental evidence that RDNAS discovers architectures with a superior balance of robustness and accuracy compared to existing baselines across multiple datasets and attack types.

**Strengths:**

1. Principled and Robust Scoring: ROSE is a key strength. Instead of using noisy gradients or standard attributions, it builds on the principled game-theoretic concept of Shapley values and thoughtfully adapts it for a high-variance, adversarial setting using robust statistical estimators. This directly addresses a known failure point in previous robust NAS works.

2. State-of-the-Art Performance: The method achieves excellent results, outperforming a wide range of baselines (e.g., AdvRush, RACL, RobNet) on standard benchmarks. For instance, on CIFAR-10, it achieves high clean accuracy while also delivering top-tier robustness against PGD and AutoAttack.

**Weaknesses:**

1. High Complexity: The overall system is complex, combining a bilevel optimization, adversarial training in the inner loop, a custom dual-branch cell with attention, and the sophisticated ROSE estimator (which itself uses Shapley values, MoM, and IQR). This complexity could make reproduction and debugging difficult.

2. New Hyperparameters: The ROSE estimator introduces its own set of hyperparameters, such as the $\beta$ parameter that balances the MoM and IQR scores, the number of MoM groups $G$, and the IQR sensitivity $\gamma$. This adds extra tuning parameters to an already complex search process.

3. Limited Search Space: The method still operates within a conventional cell-based NAS search space, where the set of operations (conv, pool, etc.) is predefined. The work is limited in the cell topology, while creating difficulty for transferring it to more complex transformer based architectures.

**Questions:**

* Regarding the ROSE Estimator:
  - The final ROSE score is a weighted sum: $Score_{e,o}^{(b)}=(1-\beta)m_{e,o}^{(b)}+\beta~v_{e,o}^{(b)}$. What is the performance impact of ablating these components? For example, what happens if $\beta=0$ (using only MoM) or $\beta=1$ (using only the IQR outlier score)?
  - How sensitive is the final discovered architecture to the choice of $\beta$? The paper suggests a range of $[0.3, 0.5]$, does the performance degrade sharply outside this range?

* Regarding the Framework and Evaluation:
  - The search uses a 7-step PGD ($PGD^7$), but the final evaluation includes stronger attacks like $PGD^{20}$, $PGD^{100}$, and AutoAttack. Does this indicate that the architecture found using a weaker attack generalizes well to stronger ones?
  - How well do the discovered architectures perform against different types of adversarial attacks not included in the main table, such as $l_2$ or $l_0$ norm-bounded attacks?

---

> ### Author Response · Authors · 2025-11-20
> **Response to Reviewer f19E**
>
> # Response to Reviewer f19E
>
> We thank the reviewer for the detailed summary and for recognizing the strengths of RDNAS, in particular the principled nature of ROSE and the strong empirical performance against both PGD and AutoAttack. Below we address the concerns about complexity, hyperparameters, search space, and the specific questions on ROSE and the evaluation protocol
>
> ### 1. On the ROSE estimator (Q1 & hyperparameter impact)
>
> > **Q1.** TQ1. What is the effect of using only MoM ($\beta = 0$=0) or only IQR ($\beta = 0$=1)? How sensitive is the final architecture to $\beta = 0$?
>
> **Response.**
> Thank you for this concrete question. To answer it directly, we refine the hyperparameter study in Appendix A.3 and **explicitly include $\beta = 0$ and $\beta = 1$** (pure MoM and pure IQR). On CIFAR-10, we set
> $\beta \in \{0, 0.2, 0.3, 0.4, 0.5, 1\}$ and report the final architecture’s clean accuracy and robust accuracy under FGSM, $PGD^{20}$, and $PGD^{100}$:
>
> Table 7: CIFAR-10 performance under different $\beta$ values. Best is **bold**, second-best is <u>underlined</u>.
>
> | $\beta$ | Clean      | FGSM          | $PGD^{20}$    | $PGD^{100}$   |
> |--------:|-----------:|--------------:|--------------:|--------------:|
> | 0.0     | 85.25%     | 60.21%    | 52.55% | 52.08% |
> | 0.2     | 85.06%     | 58.68%        | 50.54%        | 50.20%        |
> | 0.3     | **86.56%** | **60.44%**    | **52.62%**    | **52.24%**    |
> | 0.4     | 85.49% | 59.63%     | 52.22%        | 51.67%        |
> | 0.5     | 84.83%     | 58.48%        | 50.99%        | 50.47%        |
> | 1.0     | 84.45%     | 59.47%        | 52.45%        | 52.13%        |
>
> We observe that:
>
> - **Pure MoM ($\beta = 0$)** already exhibits **strong robustness**, comparable to or better than many baselines, but its clean accuracy is relatively low, so the resulting architecture is not well balanced. Our goal is to obtain architectures that are both robust and well balanced in terms of clean performance.
> - In the **intermediate mixture regime around $\beta \approx 0.3$**, we obtain the **best overall trade-off** between clean accuracy and robustness under all three attacks.
> - Over the range **$\beta \in [0, 0.5]$**, performance varies **smoothly** with $\beta$ rather than changing catastrophically; there is no “cliff” where performance collapses outside a very narrow window.
>
> These findings support our design intuition: **MoM** provides a stable scoring baseline under heavy-tailed adversarial noise, while the **IQR term** captures rare but decisive gains. The coefficient $\beta$ simply interpolates between these two behaviors. As long as $\beta$ lies in a reasonable middle range, the discovered architecture is **not overly sensitive** to its exact value. We will include this table and discussion in the revised Appendix A.3 and briefly reference it in the main text.
>
> According to the definitions of MoM and IQR, the MoM term should generally receive a higher weight than the IQR term, since IQR functions mainly as a bonus mechanism while MoM represents the stable mean statistic. For this reason, we have not yet conducted hyperparameter experiments for the more IQR-heavy settings (i.e., $\beta = 0.6, 0.7, 0.8, 0.9$).
>
> In addition, Appendix Tables 6 report the original hyperparameter experiments on $\gamma$, showing that RDNAS remains competitive across a broad grid and that our default setting lies in a stable region.

---

> > ### Author Response · Authors · 2025-11-20
> > **Response to Reviewer f19E Q2**
> >
> > ### 2.On search space and evaluation protocol (Q2–Q3)
> >
> > **Reviewer’s concerns.**
> > > (1) The method operates in a conventional cell-based CNN search space, which may limit extension to more complex transformer-style architectures.
> > > (2) The search uses 7-step PGD $PGD^{7}$, while the final evaluation considers stronger attacks such as $PGD^{20}$, $PGD^{100}$, and AutoAttack; it is unclear how this choice affects robustness and generalization.
> >
> > We deliberately adopt a **standard DARTS-like CNN cell space** and a **moderate PGD configuration during search**, in line with mainstream robust NAS (e.g., Mok et al., *AdvRush: Searching for Adversarially Robust Neural Architectures*, ICCV 2021; Dong et al., *Adversarially Robust Neural Architectures*, TPAMI 2025; Sun et al., *A3D: A Platform of Searching for Robust Neural Architectures and Efficient Adversarial Attacks*, TPAMI 2025). This design choice serves two purposes:
> >
> > 1. **Fair and focused comparison.**
> >    By keeping the operator set and cell-level search space identical to prior robust NAS work, we can attribute performance gains to **our dual-branch architecture and the ROSE scorer**, rather than to a larger or more expressive search space.
> >
> > 2. **Practical and robust evaluation.**
> >    Following the standard robust NAS protocol, we use $PGD^{7}$ in search to keep the bilevel inner loop computationally feasible, then retrain the selected architecture under a slightly stronger PGD setting with the same training schedule and hyperparameters as the robust NAS baselines (and apply this identical training setup to all compared architectures), and finally evaluate under $PGD^{20}$, $PGD^{100}$, and AutoAttack, which are strictly stronger than the search attack. **RDNAS exhibits relatively superior performance under all these attacks, indicating that the discovered architectures generalize beyond the specific $PGD^{7}$ used in search and supporting both the robustness evaluation design of RDNAS and the practical feasibility of the architectures it discovers.**
> >
> > Our framework itself is agnostic to the primitive set and norm: the dual-branch cell and ROSE only require a directed acyclic cell graph and a loss under adversarially perturbed data. Furthermore, there are currently no comparative studies on attacks of other norms; our experiments, designed to ensure fairness, are based on previous papers. We will consider your suggestions and add attack experiments for other norms in the future.

---

> > > ### Comment · Reviewer_f19E · 2025-11-27
> > >
> > > Thanks for the authors' explanation. However, I still have two major concerns not solved:
> > >
> > > 1. The authors can try transformer based architecture other than cell based archs, and there are bunch of transformer pruning and compression works for fair comparison recently.
> > >
> > > 2. For generalization to harder attacks in evaluation protocol, I appreciate the model's capability in that, but A/B test and more extenstive ablation stuties may be needed to confirm the assumption first, in stead of quickly making the conclusion without more experiment validation.
> > >
> > > In all, I decide to keep my original score. But I want to thank authors for their hard efforts of conducting hyper-parameter tuning experiments in the rebuttal.

---

> > > > ### Author Response · Authors · 2025-11-28
> > > > **Response to f19E**
> > > >
> > > > We appreciate the reviewer’s suggestion regarding Transformer-based NAS. Our goal in this paper is to study robust NAS within the widely used DARTS/NAS-Bench-201–style convolutional search spaces, where most existing robust NAS methods (e.g., AdvRush, RACL, LRNAS, TRNAS) are also defined. These works likewise focus on CNN-based cells rather than transformers and are evaluated on the standard CIFAR/Tiny-ImageNet benchmarks with PGD-style adversarial training. Extending RDNAS to transformer supernets would require a substantial redesign of the search space, operators, and robustness proxy, which we view as an interesting but orthogonal research direction. If the reviewer could point us to several representative Transformer-based Robust NAS works published in recent top-tier venues, we would be grateful and will explore including as many comparative experiments as possible within the revision timeline.

---

### Official Review · Reviewer_S3uE · 2025-10-30

**Soundness:** 4
**Presentation:** 3
**Contribution:** 3
**Rating:** 8
**Confidence:** 4

**Summary:**

This paper presents RDNAS, a robust neural architecture search framework that jointly optimizes for clean accuracy and adversarial robustness through a dual-branch cell design and a Robust Outlier-Aware Shapley Estimator (ROSE). The work is well-motivated, experimentally comprehensive, and addresses an important challenge in NAS under adversarial conditions. The empirical results across CIFAR-10/100, SVHN, and Tiny-ImageNet are strong and convincingly demonstrate the robustness–accuracy trade-off. Overall, the paper is technically sound and clearly written, along with some aspects expected to be clarified or extended to further strengthen the contribution and readability.

**Strengths:**

+ The dual-branch cell design is novel in the robust NAS community. This design cleanly separates and fuses normal and robust pathways, improving adversarial robustness without significantly enlarging the search space.

+ The proposed ROSE estimator effectively stabilizes Shapley-based operation scoring under noisy adversarial training.

+ The proposed RDNAS method achieves strong empirical performance across multiple datasets and attack settings, consistently outperforming hand-crafted and robust NAS baselines in terms of both clean and adversarial accuracy.

+ RDNAS conducts computationally efficient search with a small-sample strategy and shallow architecture, achieving high performance with low search cost.

**Weaknesses:**

+ The ROSE estimator is a good idea, but the current presentation is somewhat dense. Adding a brief intuitive explanation of how Median-of-Means and IQR filtering improve robustness, perhaps with a small ablation or visualization, would make this component clearer.

+ In Tables 1–4, consider boldfacing or underlining both the best and second-best results and adding small commentary lines like “RDNAS achieves the best balance between robustness and efficiency.” This helps readers quickly grasp the contribution.

+ Some symbols (e.g., $\alpha_{e,o}^{(b)}$, $\Delta_{e,o}^{(s,b)}$) appear with slightly inconsistent superscripts or subscripts across sections. Unifying notation between Equations (15)–(19) and the algorithm pseudocode would improve readability.

+ Briefly mention the approximate search time (e.g., 0.2 GPU Days) in the main text (not only the table). This helps readers appreciate the efficiency advantage compared to typical NAS methods.

**Questions:**

Please refer to the Weaknesses part.

---

> ### Author Response · Authors · 2025-11-20
> **Response to Reviewer S3uE**
>
> # Response to Reviewer S3uE
> We sincerely thank the reviewer for the very positive assessment of RDNAS, and for recognizing the novelty of the dual-branch cell design, the role of ROSE, and the overall empirical strength and efficiency of our method. Below we address the points raised in the “Weaknesses” section.
>
> ### Further explanation of the ROSE evaluator issue
>
> We thank the reviewer for the suggestion. In the revised version, we have updated Section 3.4 (ROSE) to provide a clearer and more intuitive explanation of how Median-of-Means and IQR jointly stabilize Shapley-based scoring under the noisy conditions of adversarial training.
>
> In brief, MoM offers a robust estimate of the typical marginal contribution by reducing the influence of heavy-tailed noise, while the IQR-based term highlights rare but significant operation effects that should not be treated as noise. The final ROSE score combines these two complementary signals through a single mixing coefficient.
>
>
> We also revisited the hyperparameter study on the mixing coefficient β and updated the ablation results in Appendix A.3, covering β ∈ {0, 0.2, 0.3, 0.4, 0.5, 1}. The revised appendix now reports the corresponding clean and robust accuracies under FGSM, PGD20, and PGD100.
>
>
> Table 7: CIFAR-10 performance under different β values. Best is **bold**, second-best is <u>underlined</u>.
>
> | β   | Clean    | FGSM      | $PGD^{20}$    | $PGD^{100}$   |
> |-----|----------|-----------|-----------|-----------|
> | 0   | 85.25%   | 60.21% | 52.55% | 52.08% |
> | 0.2 | 85.06%   | 58.68%    | 50.54%    | 50.20%    |
> | 0.3 | **86.56%** | **60.44%** | **52.62%** | **52.24%** |
> | 0.4 | *85.49%* | 59.63%    | 52.22%    | 51.67%    |
> | 0.5 | 84.83%   | 58.48%    | 50.99%    | 50.47%    |
> | 1   | 84.45%   | 59.47%    | 52.45%    | 52.13%    |
>
> ## Else Quention(Q2-Q4)
>
> We thank the reviewer for these helpful presentation suggestions. All the recommended improvements have been incorporated into the revised manuscript. Specifically, we now (i) highlight both the best and second-best results in Tables 1–4 and include brief interpretive remarks to improve readability, (ii) unify the notation of symbols such as $\alpha_{e,o}^{(b)}$ and $\Delta_{e,o}^{(s,b)}$ so that Equations (15)–(19) and the algorithm pseudocode follow a consistent indexing scheme, and (iii) explicitly report the approximate search cost (about 0.2 GPU days on CIFAR-10) in the main text in addition to the tables. We appreciate the reviewer’s suggestions and have updated the manuscript accordingly.

---

### Author Response · Authors · 2025-12-03
**General Response**

We would like to express our heartfelt thanks to all reviewers for their time and insightful feedback. In the rebuttal and revised version, we have made the following key updates and clarifications:

- **ROSE hyperparameters and robustness to β.**
We ran an extended study of the ROSE mixing coefficient β on CIFAR-10, explicitly including both β = 0 (pure MoM) and β = 1 (pure IQR). We also added a clearer explanation of how ROSE works and why it fits our robust NAS setting.
[f19E, S3uE Concern]

- **Dual-branch cell and loss design vs. single-cell baselines.**
  We clarified our original ablation to avoid confusion and, following the reviewers’ concerns, extended it by testing single- vs dual-branch cells under TRADES loss. Under TRADES, the single-branch cell becomes even less competitive than under Madry, while our dual-branch topology still clearly outperforms it. This supports that the gain mainly comes from the cell topology rather than from cherry-picking a particular loss, and we keep Madry loss as the main setting to stay comparable with prior robust NAS baselines.
  [QydH Concern]

- **DARTS-style search space, ranking consistency, and transfer.**
  We make it clear that RDNAS only borrows the DARTS cell-level operator set and CNN-style search space; we do not reuse the original gradient relaxation. We also add a NAS-Bench-201 experiment showing that ROSE gives a stable ranking in the high-performing region and finds strong architectures compared with DYNAS-SPOS.
[jtcF Concern]

- **Comparison to zero-cost proxy (ZCP) based robust NAS.**
  We spell out the difference between ZCP-style methods and RDNAS: ZCPs use cheap proxies to approximate how a robustly trained network would behave, whereas RDNAS trains architectures adversarially during the search itself. To ground this, **we add new ImageNet-1k experiments** comparing our architecture with DARTS, LRNAS, and ZCP-based robust NAS methods (CRoZe, ZCPRob, TRNAS) under FGSM and PGD-20; our model gets the best clean and robust accuracy, showing that the extra training cost really pays off.
[jtcF Concern]


- **Search space and Transformer-based architectures.**
  The current work is restricted to a DARTS-style CNN cell search space. Reviewers ask whether the method can be extended to Transformer or hybrid search spaces, and request direct comparisons to Transformer-based robust models / pruning methods. Our theme in this paper is robust architecture search for DNNs in a CNN-style framework; extending RDNAS to Transformers would require substantial redesign of the search space, operators, and robustness proxy, which we view as important future work rather than something we can do reliably within this submission. To our knowledge, there is currently no Transformer-based robust NAS method that would serve as a direct counterpart to compare with. Instead, we follow and explicitly cite the de facto setup in robust NAS, where recent methods also search CNN-style cells under adversarial training (e.g., AdvRush, ICCV 2021; Ou et al., CVPR 2024; Dong et al., TPAMI 2025; TRNAS, ICCV 2025; A3D, TPAMI 2025).

---

### Note · Authors · 2026-01-27

I have read and agree with the venue's withdrawal policy on behalf of myself and my co-authors.

---

### Meta-Review · Area_Chair_4hRE · 2026-01-06

**Summary:**

This manuscript proposes a method termed RDNAS, Robust Dual-Branch Neural Architecture Search. RDNAS is an architecture search  framework for models robust to adversarial attacks.
The idea is to use adversarial training within a DARTS-like bi-level optimization process.  A Robust Outlier-Aware Shapley Estimator (ROSE) is used to find the final architecture.
The reviews initially provide mixed scores ranging from 2 to 8, where the criticism is on the one hand on the high complexity of the proposed approach which calls for extensive ablations and on the other hand on its limitation to the search space. After the rebuttal, some questions are addressed, including several provided ablations such that the paper is a borderline case. The current restriction to CNN architectures however is a severe limitation of the work. In light of the reviews and the rebuttal, the AC therefore recommends rejection.

**Reviewer Concerns:**

The reviewers had several concerns, most importantly:
Reviewer S3uE: the reviewer initially provides a score of 8 - concerns are mostly on the writing and presentation and can be easily addressed.
Reviewer f19E: High Complexity. --> somewhat argued for
Reviewer f19E: New Hyperparameters. --> addressed by ablating on hyperparameters
Reviewer f19E: Limited Search Space --> not addressed
Reviewer jtcF: DARTS often uses smaller networks for search (fewer cells) ..... --> addressed
Reviewer jtcF: There is no information about the actual found networks. --> addressed to some extent
Reviewer jtcF: Due to the search space being based on DARTS, there’s no possibility of going beyond CNN-types. I’m not sure this should be the current NAS direction anymore. --> not addressed
Reviewer jtcF: Due to the adversarial training already during search, standard training is not possible. How can we evaluate the native robustness of the found network without needing adversarial training? --> addressed to some extent
There is no reproducibility statement.
Reviewer jtcF: A fair comparison in Table 1 would also use a similar amount of flops. --> addressed
Reviewer jtcF: Given the vast amount of NAS literature using zero-cost proxies to search for robust networks quickly, why would DARTS be necessary or can this be combined to also include the adversarial training weights? --> not addressed
Reviewer QydH: Relevance of ROSE block --> addressed
Reviewer QydH: Separate cells over different losses: I understand the reason that led the authors in designing two separate cells. But still, it is not clear to me how and why this should be better than having a single cell with a different loss, such as the TRADES ... --> addressed by an ablation study. However, I am not sure this is fully convincing since TRADES itself has hyperparameters.
Reviewer QydH:  AutoAttack as standard evaluation --> addressed

**Reviewer Scores:**

Reviewer S3uE: 8 --> probably maintained
Reviewer f19E:  2 --> likely maintained
Reviewer jtcF : 2 -->concerns and questions are addressed to some extent. The reviewer might have increase the score to 4.
Reviewer QydH: 4 --> the rebuttal addressed several important concerns.

---

### Decision · Program_Chairs · 2026-01-26

Reject